# Impact of global cooling on Early Cretaceous high $p$CO$_2$ world during the Weissert Event

Liyenne Cavalheiro [1], Thomas Wagner [2✉], Sebastian Steinig[3], Cinzia Bottini [1], Wolf Dummann[4,8], Onoriode Esegbue[2,9], Gabriele Gambacorta[5], Victor Giraldo-Gómez [1], Alexander Farnsworth [3,10], Sascha Flögel[6], Peter Hofmann[4], Daniel J. Lunt [3], Janet Rethemeyer[4], Stefano Torricelli[7] & Elisabetta Erba [1]

The Weissert Event ~133 million years ago marked a profound global cooling that punctuated the Early Cretaceous greenhouse. We present modelling, high-resolution bulk organic carbon isotopes and chronostratigraphically calibrated sea surface temperature (SSTs) based on an organic paleothermometer (the TEX$_{86}$ proxy), which capture the Weissert Event in the semi-enclosed Weddell Sea basin, offshore Antarctica (paleolatitude ~54 °S; paleowater depth ~500 meters). We document a ~3–4 °C drop in SST coinciding with the Weissert cold end, and converge the Weddell Sea data, climate simulations and available worldwide multi-proxy based temperature data towards one unifying solution providing a best-fit between all lines of evidence. The outcome confirms a 3.0 °C ( ±1.7 °C) global mean surface cooling across the Weissert Event, which translates into a ~40% drop in atmospheric $p$CO$_2$ over a period of ~700 thousand years. Consistent with geologic evidence, this $p$CO$_2$ drop favoured the potential build-up of local polar ice.

[1] Department of Earth Sciences Ardito Desio, University of Milan, Milan, Italy. [2] The Lyell Centre, Heriot–Watt University, Edinburgh, UK. [3] School of Geographical Sciences, University of Bristol, Bristol, UK. [4] Institute of Geology and Mineralogy, University of Cologne, Cologne, Germany. [5] Eni S.p.A. Natural Resources–Geology and Geophysics Research and Technological Innovation, San Donato Milanese, Milan, Italy. [6] GEOMAR Helmholtz Centre for Ocean Research, Kiel, Germany. [7] Eni S.p.A. Natural Resources–Sedimentology, Stratigraphy and Petrography Department, San Donato Milanese, Milan, Italy. [8] Present address: Institute of Geosciences, Goethe-University Frankfurt, Frankfurt am Main, Germany. [9] Present address: School of Natural and Environmental Science, Newcastle University, Newcastle, UK. [10] Present address: State Key Laboratory of Tibetan Plateau Earth System, Resources and Environment (TPESRE), Institute of Tibetan Plateau Research, Chinese Academy of Sciences, Beijing 100101, China. ✉email: t.wagner@hw.ac.uk

The greenhouse world of the Mesozoic–Paleogene is a primary target for understanding the behaviour of the Earth system during periods of extreme warmth and its response to recurrent episodes of short-term warming[1–4] and cooling[5–7], including the possible build-up of polar ice[8]. The Late Valanginian (Early Cretaceous) Weissert Event, calibrated in the uppermost part of magnetic chron CM12 through the upper part of magnetic chron CM11[9] (~133.9–132.6 million years ago; updated chronological framework in this study) has large potential to shed light on the climate-carbon-ice relationships at elevated but not very high atmospheric $pCO_2$ levels (~500–1700 ppm[10]). The Weissert perturbation is globally documented in the sedimentary record by a positive (+1.5‰) carbon isotope excursion (CIE) observed both in organic and inorganic records[9]. The Weissert CIE, like other Cretaceous climate perturbations[5–7], is associated with extended volcanism (Paraná-Etendeka Large Igneous Province[9]), a carbonate crisis in pelagic and neritic environments, local to regional enhanced productivity, and oceanic dysoxia/ anoxia that likely boosted marine organic carbon ($C_{org}$) burial, in combination triggering atmospheric $pCO_2$ drawdown and global cooling[9,11–16]. However, there are only few proxy records[6,15–18] that document the complete temperature evolution across the Weissert Event. Even less is known about absolute atmospheric $pCO_2$ levels and the extent of transient polar ice buildup. Many temperature proxy records are inconsistent in their interpretation due to contrasting ocean temperature reconstructions based on $TEX_{86}$[17], oxygen isotopes[16,18] and Mg/Ca palaeothermometers[15], no systematic comparison with independent evidence, such as climate model simulations, and a lack of alignment against unified chronological frameworks. Furthermore, Valanginian stratigraphic sections from the climatically sensitive sub-polar/polar regions are particularly rare[16,17,19,20], fostering the debate regarding whether the Weissert cooling led to sub-polar glaciation[8,19,21], as suggested by the occurrence of Late Valanginian glendonites, dropstones, tillites, and ice-rafted debris both in sub-Arctic and sub-Antarctic regions (see[16]).

Here we present high-resolution data from a ~14 metres long, calcium carbonate-rich and finely laminated black shale section (78.61–93 metres below seafloor (mbsf); cores 113–692–10R and 113–692–12R) from Ocean Drilling Program (ODP) Site 692 (Leg 113)) in the Weddell Sea (East Antarctica)[22]. Sample material was provided by GCR—Gulf Coast Repository Texas A&M University, College Station, Texas, under request 067321IODP. Site 692 was located in a semi-enclosed shelf basin with a paleowater depth of ~500 m and a paleolatitude of ~54 °S[22]. The Valanginian paleogeography with the location of ODP 692 is shown in Fig. 1. The study section covers ~5 Myr of climate history from the Late Berriasian–Early Valanginian (~135 Myr; updated chronostratigraphy in Fig. 2) to the Early Hauterivian (~130.5 Myr). A total of 81 samples with an average time resolution of around ~25–50 kyr (Fig. 2) were measured for total organic carbon (TOC) concentration and bulk organic carbon isotopes ($\delta^{13}C_{org}$), resolving the entire Weissert Event. We also present a detailed ($N = 48$ samples) sea surface temperature (SST) profile derived from the organic glycerol dialkyl glycerol tetraether (GDGT) paleothermometry $TEX_{86}$[23]. 65 samples were used for calcareous nannofossil investigation to refine the original biostratigraphy[22] and 31 samples were investigated for benthic foraminifera assemblages to reconstruct paleowater depths. These data enable us to constrain the SST evolution of the Early Cretaceous Weddell Sea within a refined chronological framework. In order to contextualize the local Weddell Sea results in the global climate evolution of the Weissert Event, we supplement these data with other available Valanginian proxy temperature evidence from mid-latitude and tropical ocean basins, including $TEX_{86}$, oxygen isotopes, clumped isotopes and Mg/Ca measurements. In

particular, we assess uncertainties in absolute $TEX_{86}$-derived temperatures and possible regional changes in the $TEX_{86}$-SST relation for young, restricted ocean basins[24]. The temperature data from all study sites are aligned against a refined chronological framework, defined by the Weissert CIE. By comparing this chronologically aligned multi-proxy ocean temperature database to global paleoclimate modelling results we present a best-fit approach to resolve the Weissert carbon-climate perturbation. It is important to note that this best fit approach does not prefer any specific temperature calibration or location, however, it identifies techniques and study sites that are more consistent with available global observations and global modelling systematics than others. Based on this outcome we estimate the global mean surface temperatures and associated model $pCO_2$ concentrations for the different phases of the perturbation. This unified solution is then used to explore the probability of ice build-up around Antarctica as a result of atmospheric $CO_2$ drawdown during the Weissert Event.

## Results

**Weissert Event chrono- and chemostratigraphy at Site 692.** We started by combining calcareous nannofossil data and chemostratigraphy ($\delta^{13}C_{org}$) to identify the Weissert Event based on its biostratigraphically constrained positive CIE[9] (Fig. 2, Supplementary Figs. 1 and 2 and 'Methods' section). Consistent with other study sites[9,16,25], we find an average amplitude of about 1.5‰ (~ −31.5 to −30‰) and identify the onset of the event (A in Fig. 2), and two distinct maxima (B and C in Fig. 2), suggesting that the Weddell Sea record encompasses the full magnitude and duration of the Weissert CIE. The onset of the event is placed at the base of the positive CIE at ~90.5 mbsf (A in Fig. 2) and the end at ~83.5 mbsf (top of the CIE plateau, C in Fig. 2) following the original definition[9]. The recovery period, marking the return of isotope trends towards pre-perturbation levels, follows point C in Fig. 2, the end of the Weissert perturbation.

**Contrasting $TEX_{86}$-derived temperatures across the Weissert Event.** Our $TEX_{86}$ profile from Site 692 documents variable values between 0.77 and 0.63 (Eq. 1 and 'Methods' section) over a time period of approximately 750 kyr (Fig. 2). Independent of the choice of $TEX_{86}$-SST calibration, the record shows the highest (warmest) $TEX_{86}$ values in the Late Berriasian–Early Valanginian interval and during the initial warm phase of the Weissert Event (~0.74 in A–B in Fig. 2). Two minima (coolest temperatures, ~0.66) in the $TEX_{86}$ record coincide with the Weissert CIE end (C in Fig. 2) and the Valanginian/Hauterivian boundary, detached from the Weissert Event. The two cooling intervals are separated by a (<900 kyr) warming interlude. The main $TEX_{86}$ decrease (B–C in Fig. 2) of almost 0.1 (translating in a 3–4 °C cooling, discussion below in $TEX_{86}$ calibration section) is significantly larger than the analytical uncertainty of $TEX_{86}$ analysis (0.004 following published[26] analytical methodology; see 'Methods' section).

In order to assess whether the local Weddell Sea cooling represents a global climate signal, we used a published compilation of Cretaceous $TEX_{86}$ data[27], focusing only on Valanginian sites (e.g. DSDP Sites 603[17] and 534[17], ODP Site 766[17], and including also 5 published sample data from the stratigraphic study interval at ODP Site 692[17]). Palaeographic locations (Fig. 1) and a summary overview of these datasets are presented in Fig. 3, following our revised and updated bio-chemostratigraphic framework of the Weissert Event. The $TEX_{86}$-derived cooling trend at Site 692 is in stark contrast with records of time-equivalent strata at sub-tropical sites (603 and 534; ~18–25 °N) from the proto-North Atlantic[17], which show stable

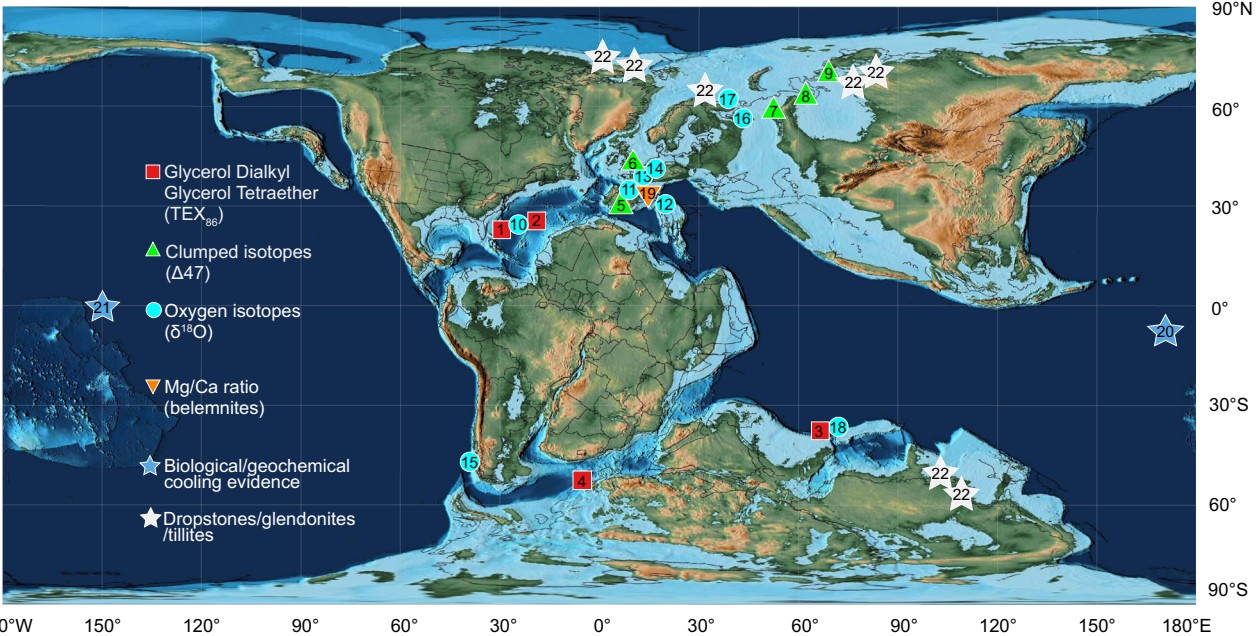

**Fig. 1 Paleogeographic reconstruction of the Valanginian–Early Hauterivian (~133 Myr).** Paleogeographic map by C.R. Scotese, PALEOMAP project[64] modified in this work to show the location of available Valanginian–Early Hauterivian multi-proxy-based temperature records (e.g. TEX$_{86}$, oxygen carbonate clumped isotopes, oxygen isotopes and belemnite Mg/Ca ratios) from Deep Sea Drilling Project/Ocean Drilling Project (DSDP/ODP) Sites and other worldwide sections: (1) 534, (2) 603, and (3) 766[17]; (4) 692[this study and 17]; (5) Caravaca, (6) Speeton, (7) Izmha, (8) Yatria and (9) Boyarka[20]; (10) Proto North Atlantic compilation[16]; (11) South East (SE) France compilation[6, 18]; (12) Northwestern Tethys compilation[18]; (13) Southern Boreal compilation[18]; (17) Festningen and Janusfjellet[19]; (18) 765[15]; (19) Vocontian Trough compilation[15]. Biological and geochemical cooling evidence in sub-tropical oceans (blue stars) from calcareous nannofossils at (20) Site 1049[9] and from steryl ethers at (21) Site 1213[33]; (22) compilation of Late Valanginian geologic cooling evidence in sub-polar regions (white stars) from dropstones, glendonites and tillites[16]. Red box symbols indicate TEX$_{86}$ data, green triangles clumped isotopes, turquoise circles oxygen isotopes, orange down-pointing triangles Mg/Ca ratio (belemnite).

and high TEX$_{86}$ values across the perturbation (~0.9 and 0.93 between the stratigraphic interval identifying the Weissert CIE). However, the $\delta^{13}C_{org}$ data from Site 766 do not unequivocally identify the CIE, and magnetostratigraphy[17] suggests an age slightly younger than the end of the CIE (i.e., the base of this record starts in the uppermost CM11 polarity chron, and thus it sits slightly above the end of the Weissert Event; Fig. 3).

We conclude that this study presents the first robust TEX$_{86}$-based evidence of a cooling episode coinciding with the end of the Weissert CIE that contrasts the stable SSTs in the sub-tropical North Atlantic[17]. We will resolve this apparent contrast by comparing the TEX$_{86}$-based results with a global compilation of other independent temperature proxies in the following section.

**Integrating global ocean temperatures and cooling evidence.** In order to explore the global nature of the Weissert Event, we evaluate available upper ocean temperature reconstructions from other geochemical proxies, such as carbonate clumped isotopes[20], oxygen isotopes[6,16,18,19,28] and belemnite Mg/Ca ratios[15] positioned in our revised and updated bio-chemostratigraphic framework (Fig. 3). A summary table of all available temperature records encompassing the Weissert CIE is reported in Supplementary Table 1. Oxygen isotopes on belemnites and/or benthic foraminifera[6,18], consistent with Mg/Ca measurements on belemnites[15], document a ~1–2 °C cooling at the end of the Weissert Event in South East France (~26 °N) and a cooling of up to 4 °C in the southern and arctic part of the Boreal Realm (~38–65 °N)[18]. Notably, our detailed stratigraphic re-analysis of these datasets also indicates that the cooling coinciding with the end of the Weissert CIE is followed by an intermittent warming, and a second minor cooling period that continues in the earliest Hauterivian (Fig. 3); a pattern also noted at the Weddell Sea study

site. Moreover, a recent overview of oxygen isotopes on bulk carbonate in the proto-North Atlantic and in the North-West and Southern Tethys[16] also documents a cooling in both hemispheres coinciding with the end of the Weissert Event. We note a strong correlation of the cooling trend documented at Site 692 with the only other available detailed record in the Southern Hemisphere at Site 765 (see ref. [16]). The stratigraphic resolution of available Valanginian clumped isotope data[20] impedes identification of the Weissert CIE but according to biostratigraphy document average temperature values of the warm initial phase of the Weissert CIE and/or of the interval shortly preceding the onset of the perturbation (Fig. 3). Within chronological uncertainties, we conclude that the temperature trends at Site 692 are in agreement with these independent global cooling signals. The absolute cooling of about 3–4 °C in the Weddell Sea is larger than for most low-to-mid-latitude sites, which is consistent with the concept of polar amplification[29], leading to a more pronounced high-latitude temperature change. Finally, we consider indirect and geological evidence supporting cooler conditions during the Early Cretaceous. Limited palynoflora data at Site 692[30] suggest the presence of a cool temperate forest on the Antarctic continent, with high moisture levels and strong seasonality with temperature below freezing. From the analysis of a qualitative dataset[30], we observe that the sample at 84.95 mbsf yields the most diverse palynofloral assemblage. Relying on previous work botanical affinities and ecological preferences[31], we postulate that this assemblage, which comprises ferns, Lycopods, Pteridosperms and conifers, is expression of cooler and more humid climatic conditions compared to assemblages above and below. The occurrence of *Podocarpidites*, a typical boreal floral element with broad temperature range but preference for cool and humid conditions[32] is worth noting as this taxon is absent from samples above and

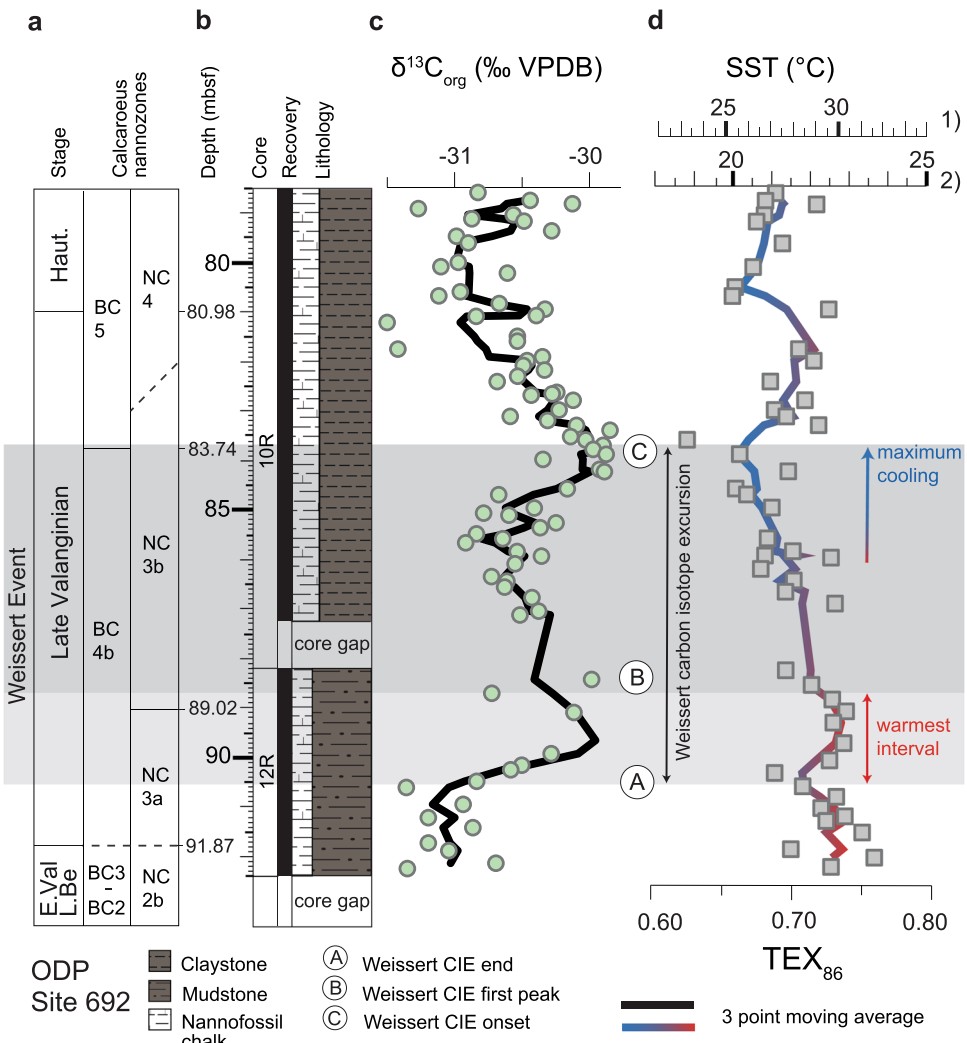

**Fig. 2 Carbon isotopes and sea surface temperatures (SSTs) of cores 10R and 12R at ODP Site 692. a** Stage and calcareous nannofossil zonation (Boreal Realm, BC[65] zonation and Tethys, NC[66] zonation). **b** Depth in metres below sea floor (mbsf), cores (12R and 10R), core recovery, and lithology. **c** Stable carbon isotope record measured on bulk organic matter ($\delta^{13}C_{org}$ with green circles and a black curve reporting the calculated three point moving average) and identification of the Weissert positive carbon isotope excursion (CIE) segments (A, Weissert CIE onset; B, Weissert CIE first peak; C, Weissert CIE end). **d** Organic TEX$_{86}$ paleothermometry (grey boxes and a shaded red (warmer values)—blue (colder values) curve reporting the calculated three point moving average) and the calibration to SSTs in degree °C showing (1) BAYSPAR and (2) restricted basin calibrations, representing maximum and minimum temperature estimates based on TEX$_{86}$, respectively.

below 84.95 mbsf. Interestingly, sample 84.95 mbsf is the closest to TEX$_{86}$ minimum (i.e., coolest temperature) recorded in this study (at 83.74 mbsf) and coinciding with the Weissert Event end (C in Fig. 2). Moreover, latest Valanginian Steryl ethers (ODP 1213, Shatsky Rise, Central Pacific[33]) and changes in calcareous nannofossil assemblages (ODP 1049, Nadezhda Basin, Western Pacific[11,34]) have been reported in sub-tropical sites, suggesting biological evidence of cooler waters in these regions. Furthermore, several findings of latest Valanginian glendonites, dropstones, tillites deposits and ice-rafted debris both in the Arctic and Antarctic sub-polar regions (>50 °N/S) (see ref. [16] compilation and locations in Fig. 1) are documented. However, the supporting evidence from sub-polar regions[21,35,36] has to be interpreted with caution due to large stratigraphic uncertainties covering the Berriasian to the Early Valanginian, but not specifically the target period of the Weissert Event. Despite stratigraphic and calibration uncertainties from all available geological and proxy evidence we note that the parallel temperature response of both hemispheres centered around the Weissert

Event is consistent and argues for a global process, with fluctuations in atmospheric $p$CO$_2$ driving temperature.

**Absolute TEX$_{86}$-derived sea surface temperatures**. A variety of calibrations have been proposed to convert measured TEX$_{86}$ ratios to upper ocean temperatures (see ref. [27] and 'Methods'). The aim of this study is neither to discuss advantages and disadvantages nor the ecological and statistical justification of individual approaches, but rather to apply multiple calibrations to generate a plausible range of absolute sea surface temperatures for each site. We will then use our multi-proxy compilation and Valanginian climate model simulations to constrain this proxy uncertainty and derive climatic implications by a best-fit approach between all datasets.

In our analysis, the BAYSPAR model[37,38] provides the upper end estimate of absolute TEX$_{86}$-derived ocean temperatures (i.e., max-TEX$_{86}$). We also report the outcomes from all calibrations in the Supplementary Data Appendix. Applying the default BAYSPAR Deep-Time settings, SSTs in the Weddell Sea record

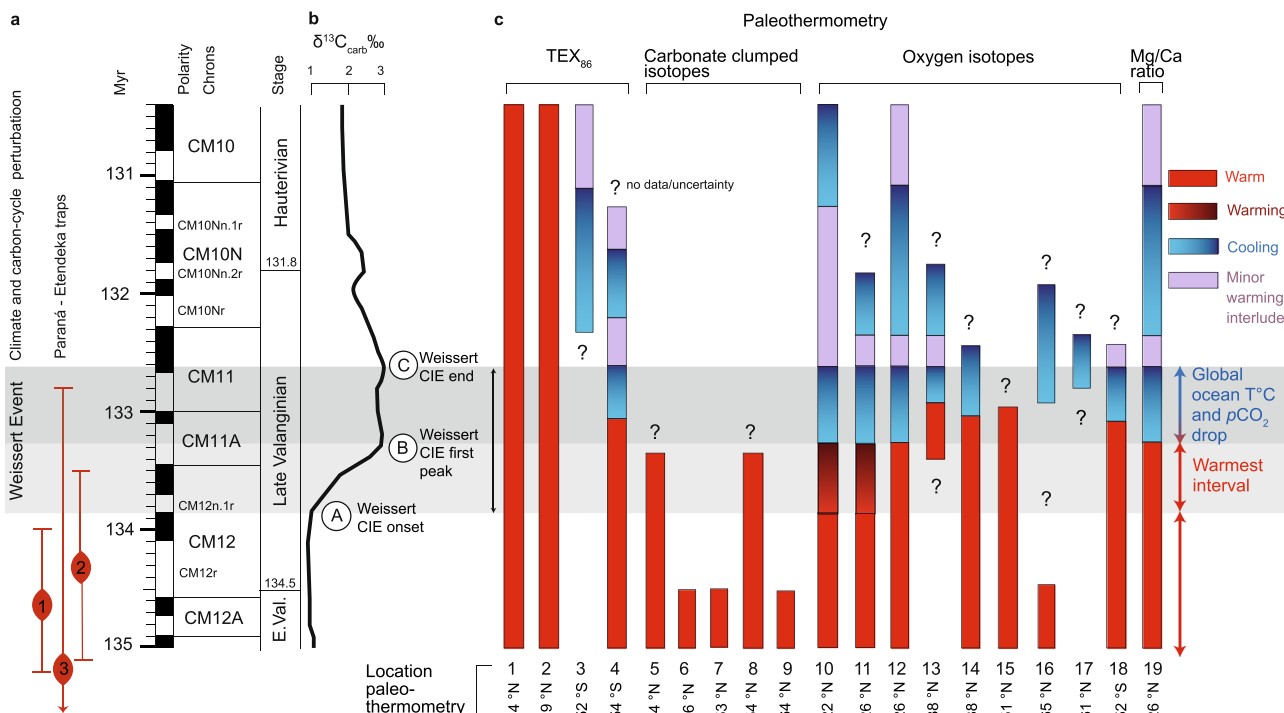

**Fig. 3 Chronological framework of the Weissert Event and evolution of worldwide multi-proxy ocean temperature trends. a** Chronostratigraphy of climate and carbon-cycle (Weissert) perturbation event and Paraná-Etendeka timing after (1)[67], (2)[68] and (3)[69]; numerical ages in Myr are based on boundary ages of polarity chrons[57] revised in this study for the position of the Valanginian/Hauterivian boundary[56]. **b** Identification of the Weissert positive carbon isotope excursion (CIE) is based on the reference carbon isotope record measured on bulk carbonate ($\delta^{13}C_{carb}$)[25] calibrated in the Southern Alps in the uppermost part of magnetic chron CM12 (A, CIE onset) and in the upper part of magnetic chron CM11 (C, CIE end)[9]. **c** A detailed correlation to available global multi-proxy-based ocean temperature records is presented by coloured bars (e.g., red (relatively stable warm interval), graded red to darker red (warming interval), blue to darker blue (cooling interval), and lilac (minor warming interlude)), which graphically represent the reconstructed ocean temperature trends organized by type of proxy (e.g., TEX$_{86}$, carbonate clumped isotopes, oxygen isotopes and Mg/ca ratio paleothermometry). Note that numbered section/site locations of the reported proxy compilation are shown in Fig. 1.

decrease from ~30 °C (warm A–B interval in Fig. 2) to a minimum of ~25.6 °C at the cold CIE end (C in Fig. 2). Importantly, results for the logarithmic TEX$_{86}$H calibration[39] (Supplementary Data Appendix) are similar to the BAYSPAR method for the Weddell Sea TEX$_{86}$ range of 0.63 to 0.77. We compare these results to published Valanginian BAYSPAR-derived SSTs[27] (Supplementary Table 1). At Site 766, the temperatures calculated from the 8 lowermost data points (~25 °C[27]) possibly represent the coolest temperatures at the end of the Weissert Event (see Fig. 3). The stable BAYSPAR SSTs in the subtropical proto-North Atlantic exceed 40 °C[27] across the perturbation. These high surface temperatures seem irreconcilable with Cretaceous climate model results[20,24], and foster the debate about the maximum heat stress tolerated by Cretaceous plants and mammals[40].

An alternative explanation for these high TEX$_{86}$ ratios may come from a distinct sedimentary isoprenoid-GDGT (i-GDGT) distribution leading to a different TEX$_{86}$-temperature relation in young and restricted Mesozoic ocean basins[24]. It is important to note that Site 766 offshore Australia represents an open marine setting with unrestricted connection to the Early Cretaceous ocean[41]. In contrast, proto-North Atlantic Sites 534 and 603 represent more restricted settings in a young and evolving ocean basin. As recently proposed[24], these special environmental conditions might have enhanced the contribution of i-GDGTs from deep-dwelling Thaumarchaeota populations, similar to observations from the modern Mediterranean and Red Sea[42] that lead to a regional warm bias in TEX$_{86}$-derived temperatures. Like the proto-North Atlantic, Site 692 in the emerging Weddell

Sea also represents a highly restricted depositional environment, although with different climate conditions[16,17,30], i.e., arid and warm versus humid and rather cool, respectively. Importantly, a consistent warm bias in TEX$_{86}$-SSTs has also been found in Pleistocene sapropels from the Mediterranean Sea[43], a highly stratified depositional environment more comparable to the Valanginian Weddell Sea (Supplementary 'Anomalous GDGT distributions' section). We also find similarities in the i-GDGT distributions of Site 692 and present-day samples from the Mediterranean Sea (Supplementary Fig. 4). Our benthic for-aminiferal data from Site 692 further indicate an intermediate-deep basin with an outer neritic to upper bathyal paleowater depth up to 500 m (Supplementary 'Benthic foraminifera' section). Such a paleowater depth would allow enhanced contribution of deeper water Thaumarchaeota communities producing the distinct i-GDGT distribution recorded in the sediments. Moreover, we document a high GDGT-2/GDGT-3 ratio >5 (on average 5.7; Supplementary Data Appendix) in the study samples, which also corroborates a contribution from Archaea living in the deeper water column[44]. Further supporting evidence for a restricted Early Cretaceous Weddell Basin with persistent ocean stratification comes from plate-tectonic recon-structions that show early stages of basin evolution with closed or limited ocean gateways[22] and very high TOC concentrations (range 3.2–12.5%; Supplementary Data Appendix) in black shales at Site 692.

For Sites 603, 534, and 692, the restricted basin calibration accounting for the regional TEX$_{86}$-temperature relation[42] (Eq. 2) yields overall ~6–14 °C cooler SSTs compared to the BAYSPAR

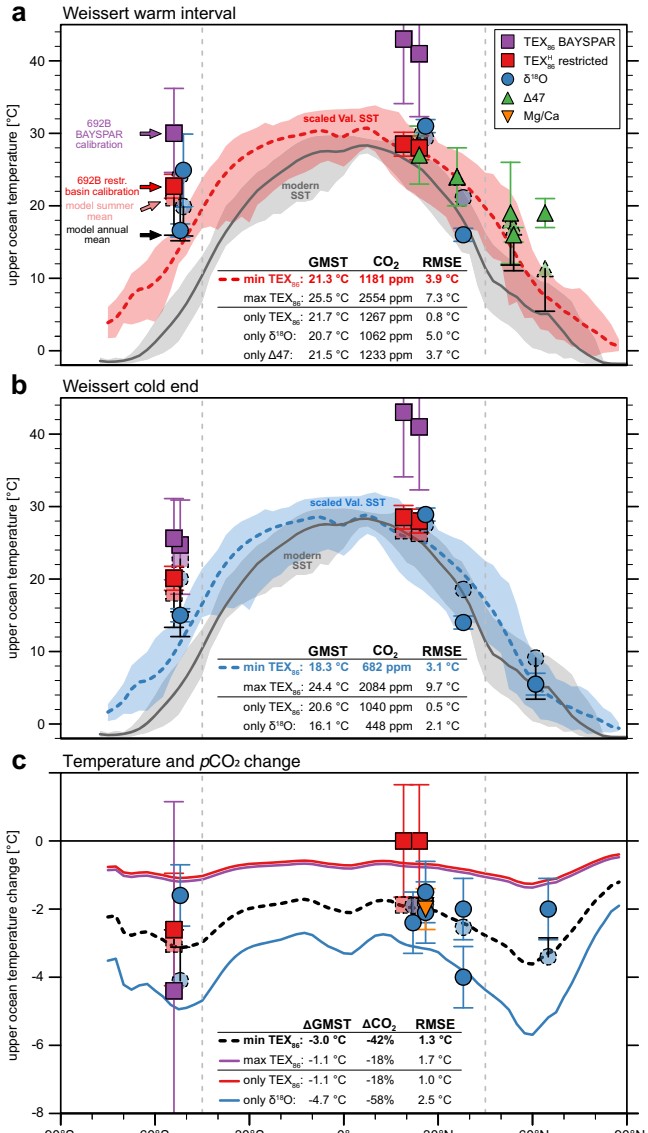

**Fig. 4 Best-fit model-proxy comparison of upper ocean temperatures and model CO₂ estimates.** Estimates of global mean surface temperatures (GMSTs) and associated model $CO_2$ levels are calculated for the whole dataset as well as for three different subsampling experiments including only single proxy techniques (only-$TEX_{86}$, only-$\delta^{18}O$ and only-$\Delta 47$). In addition, $TEX_{86}$-derived temperatures are calculated from either the calibration for modern deep, restricted basins (min-$TEX_{86}$) or the BAYSPAR calibration (max-$TEX_{86}$) for DSDP/ODP sites 534, 603 and 692. Error bars document 90% confidence intervals for the $TEX_{86}$ calibration and uncertainties for other proxies as reported in Supplementary Table 1. **a** Model SSTs are linearly scaled to the best-fit $CO_2$ estimate of 1181 ppm during the Weissert warm interval (see Supplementary Equation 5). Root-mean-square-error (RMSE) between scaled model temperatures and local proxy data are discussed in the Supplementary 'Global mean surface temperature and $p$CO2 estimates' section. Shading around the zonal mean shows the range of annual mean SSTs at each latitude, for modern[70] (grey) and Valanginian (red and blue). Dashed symbols represent simulated annual (summer) mean SSTs at low (high) latitude proxy locations for the derived mean $CO_2$ concentrations. Black bars show the difference between the warmest 3-months and annual mean temperature for high-latitude sites. **b** Model SSTs linearly scaled to the mean $CO_2$ estimates of 682 ppm during the Weissert cold end. **c** Model temperature and $p$CO₂ change from the Weissert warm interval to Weissert cold end.

results. Even though the exact ecological justification for the distinct i-GDGT distributions remains uncertain[24], we interpret the results as a possible lower end estimate (i.e., min-$TEX_{86}$) of absolute SSTs in these restricted environments. Notably, a recent study has also demonstrated that semi-enclosed basins document systematically offset from zonal mean temperatures, and are unusually warm and seasonal compared to the majority of open ocean locations situated at the same latitudes, especially in the higher latitudes[45]. We next validate the possible temperature range against our multi-proxy compilation and independent climate model results.

In summary, absolute temperatures for the Weddell Sea decrease from ~22.7/30 °C (min-/max-$TEX_{86}$) for the Weissert warm interval to a minimum of ~20.1/25.6 °C (min-/max-$TEX_{86}$) at the cold CIE end. It is important to note that while absolute temperature reconstructions change significantly with the applied calibration model, the relative cooling of ~3–4 °C observed at Site 692 is similar for all calibrations.

**Global mean surface temperature and $p$CO₂ estimates.** We estimate the global mean surface temperature (GMST) evolution across the Weissert Event by comparing our proxy compilation (Fig. 3; Supplementary Table 1) with model-simulated SSTs. This approach assumes that the global mean surface temperature scales linearly with local SSTs and that this scaling factor can be estimated from two model simulations at ×2 and ×4 preindustrial atmospheric $p$CO₂ concentrations (e.g., 560 and 1120 ppm)[46]. This approach allows us to derive an independent GMST estimate for each proxy site via a single transfer function (Supplementary Equation 3). We then calculate average GMSTs across all sites for both the Weissert warm interval between the Weissert CIE onset to the first peak CIE (warm A–B interval in Fig. 3) and the cold Weissert CIE end (C in Fig. 3). We further use the associated standard errors across all proxy sites as a measure of overall consistency between different proxy techniques and the model results.

Combining our multi-proxy compilation, the lower-end estimates of $TEX_{86}$-derived SSTs (min-$TEX_{86}$), and annual mean model temperatures we estimate GMSTs of 24.4 °C (±1.8 °C) and 21.5 °C (±1.6 °C) for the Weissert warm interval and cold CIE end, respectively (Supplementary Table 2). Assuming a linear climate sensitivity this indicates a model-derived atmospheric $p$CO₂ concentration above 2000 ppm for the Weissert warm interval (Supplementary Fig. 6 and Equation 6). Even though these estimates depend on the climate sensitivity of the model, with higher climate sensitivity resulting in lower $CO_2$ values and vice-versa, they clearly exceed the plausible range of available Valanginian $p$CO₂ reconstructions (~500–1700 ppm;[10]). These high $p$CO₂ estimates are related to very warm GMSTs derived from mid to high latitude sites of both hemispheres (Supplementary Fig. 5). High-latitude proxy reconstructions significantly warmer than climate model temperatures are a long-standing challenge for the model-data comparison of past greenhouse climates, including the Valanginian[20]. A potential reason for this mismatch is a strong seasonality in available sunlight and food web dynamics, which has the potential to skew higher latitude proxy reconstructions towards seasonal rather than annual mean temperatures (e.g., refs. [47–49]). We explore this hypothesis by repeating the analysis with simulated warmest 3-month mean instead of annual mean temperatures for mid to high latitude sites poleward of 45 °. The assumption of a summer bias in high-latitude proxy reconstructions reduces absolute GMST estimates to 21.3 °C (±1.2 °C) for the Weissert warm interval and 18.3 °C (±1.1 °C) for the Weissert cold end, a reduction of about 3 °C for

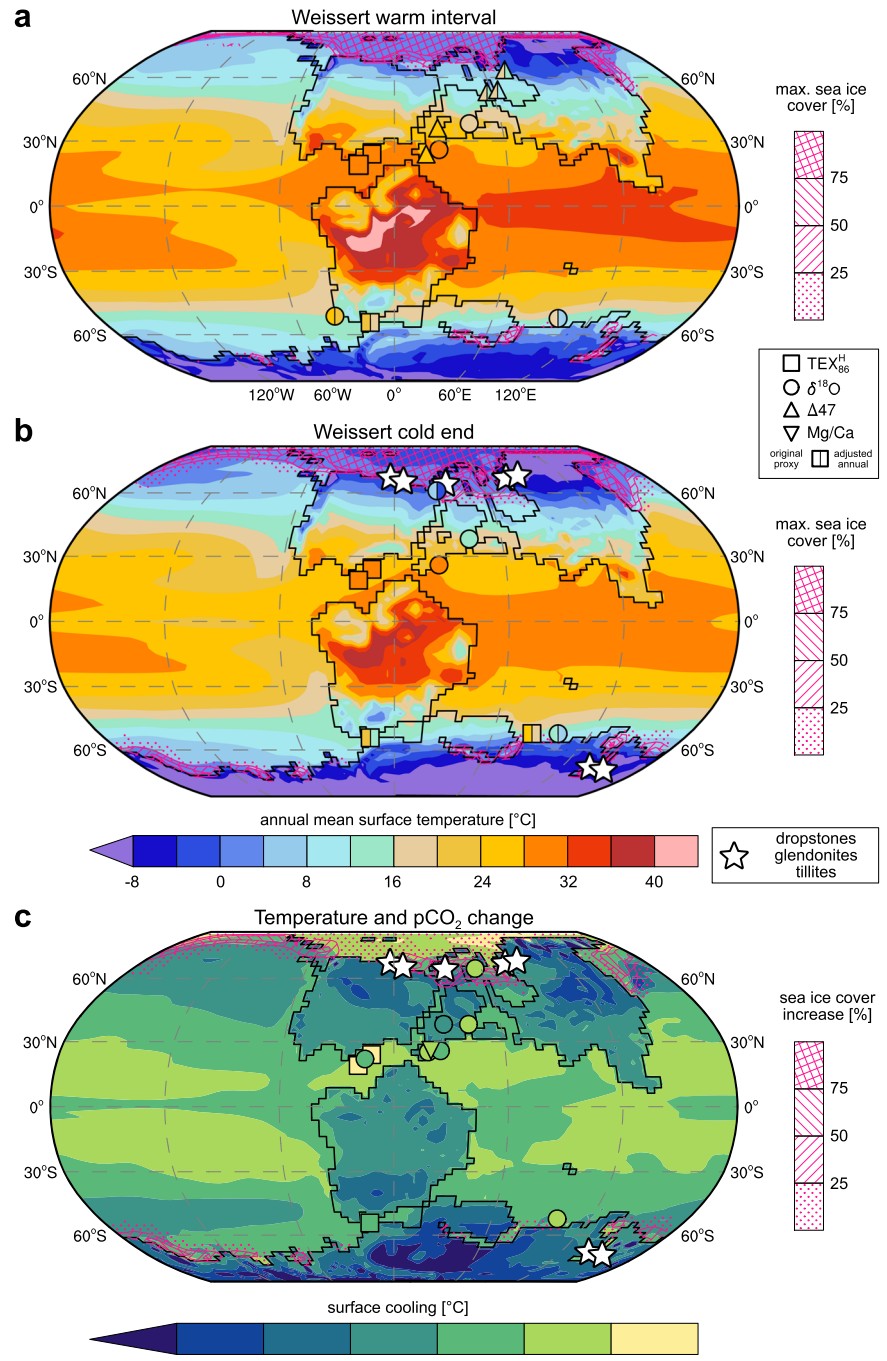

**Fig. 5 Simulated and reconstructed annual mean surface temperatures and cooling across the Weissert Event. a** Model fields combine surface air temperatures over land with sea surface temperatures (SSTs) over the ocean and are linearly scaled to the calculated mean $p\mathrm{CO_2}$ estimates of 1181 ppm during the Weissert warm interval (Supplementary Equation 5). Pink contours show associated maximum monthly mean sea ice concentrations. Symbols represent available proxy information. High-latitude sites (poleward of 45 °) also show approximated annual mean temperatures where the model-estimated difference between summer and annual mean values is subtracted from the original proxy temperatures to correct for a possible seasonal warm bias and to facilitate the model-data comparison. **b** Annual mean surface temperatures during the Weissert cold end scaled to the mean $p\mathrm{CO_2}$ estimate of 682 ppm. **c** Estimated surface cooling from the Weissert warm interval to the Weissert cold end, i.e. panel **a** minus **b**, with an associated $p\mathrm{CO2}$ drop of ~40%.

both periods (Supplementary Table 2). The reduced standard errors compared to the annual mean analysis document an overall more consistent distribution in simulated and reconstructed surface temperatures. Associated $p\mathrm{CO_2}$ concentrations under this assumption are ~1180 ppm (940–1480 ppm) and ~680 ppm (550–840 ppm) for the Weissert warm interval and cold CIE end, respectively (Fig. 4), and therefore within the range of independent $p\mathrm{CO_2}$ reconstructions[10]. Our results indicate a drop in GMST of 3.0 °C (±1.7 °C) towards the Weissert CIE end (Fig. 4c). This global cooling and the corresponding ~40% reduction in $p\mathrm{CO_2}$ is independent of any seasonal bias assumption (Supplementary Fig. 6), and in good agreement with proxy estimates from carbon isotopes on terrestrial plant records[13].

We further repeat our analysis for four different variations of the proxy data to assess the sensitivity of our results to the selected proxy techniques and temperature calibrations. The max-$TEX_{86}$ data-set (BAYSPAR) increases local proxy temperatures by ~6–14 °C in the proto-Weddell Sea and proto-North Atlantic sites and therefore overall GMST estimates (Supplementary Table 2). The corresponding model $pCO_2$ concentration of more than 2500 ppm for the Weissert warm interval by far exceeds the upper end of available reconstructions and results in overall higher root-mean-square-errors in the model-data comparison (Fig. 4). The increased standard errors of the mean for max-$TEX_{86}$ compared to min-$TEX_{86}$ (Supplementary Table 2) indicate an overall worse fit between BAYSPAR-derived GMST estimates and those based on other proxy methods. We, therefore, observe that the highest degree of consistency between model temperatures and our proxy compilation is achieved by application of a regional $TEX_{86}$ calibration for restricted Mesozoic ocean basins for selected sites in combination with a seasonal warm bias in high-latitude SST proxies. Only this specific combination permits a model-data congruence within the plausible range of $pCO_2$ reconstructions for the Valanginian[10,50,51] and will be further discussed in the next section.

**High-latitude cooling and ice formation during the end of the Weissert Event**. The cold Weissert CIE end is associated with a drop in the GMST of 3.0 °C (±1.7 °C) compared to the Weissert warm interval, but the modelling results indicate large contrasts in the regional cooling patterns (Fig. 5c). Minimum cooling of 1–2 °C is simulated for the subtropics, while it exceeds 6 °C in polar, continental regions. The reduced sensitivity of proto-Arctic Ocean temperatures can be explained by already near-freezing temperatures during the Weissert warm onset (Fig. 5a). The model supports the possibility for seasonal sea ice formation in the Arctic and around Antarctica for both Weissert $pCO_2$ scenarios. The seasonal sea ice coverage however expands during the cool end of the event, with largest increases around the Arctic. The latitudes of maximum simulated cooling and sea ice extension (~50–70 ° in both hemispheres) coincide with the location of upper Valanginian dropstones, glendonites and tillites[16] (Fig. 5c), indicating cooler surface waters, sea ice and drifting ice conditions. We quantify net snow mass accumulation rates as a first-order approximation for the potential of ice formation on Antarctica (Supplementary Fig. 7). The total snow mass surviving the Antarctic summer increases by ~58% in the ×2 $pCO_2$ simulation (~cold Weissert CIE end) compared to the ×4 $pCO_2$ results (~warm Weissert interval). Areas of potential land ice formation are restricted to southern and western coastal, high-elevation (>2000 m) regions. Previous modelling results support a $pCO_2$ threshold for the onset of Antarctic glaciation between ×2 to ×4 pre-industrial levels for the Early Cretaceous[52] and the Cenozoic[53], even though absolute numbers depend on the climate model[54]. Within model and paleogeographic uncertainties, our simulations support small scale continental glaciation during the Valanginian, consistent with geological observations. Moreover, calcareous nannofossil data from Site 692 (this study) confirm coeval braarudosphaerid-enrichments documented offshore Antarctica (ODP Sites 766 and 765[55]; and Supplementary 'Paleoecology of braarudosphaerids' section), and at high latitudes in the northern hemisphere (e.g., ref. [34], Supplementary 'Paleoecology of braarudosphaerids' section) in the latest Valanginian–Early Hauterivian (Supplementary Fig. 3). These braarudosphaerid-enrichments thus suggest temporary salinity lowering possibly triggered by discharges of fresh, deglacial melt water during the warming interlude following the cold Weissert CIE end, in high latitude and subpolar regions. Unfortunately, there are significant discrepancies in recent Early Cretaceous

global sea level changes and variations in ocean water salinity in high-latitude basins from oxygen isotopes (see ref. [16]).

## Discussion

Our findings from geological evidence, proxy records and climate model simulations provide a calibrated context of global climate perturbation for the ~2 million years history of the Early Cretaceous Weissert Event. This study documents the first $TEX_{86}$-based evidence of a relevant cooling episode coinciding with the end of the Weissert perturbation in the sub-polar proto-Weddell Sea. The observed cooling (3–4 °C) is consistent with global signals based on different types of SST proxies that overall document a more pronounced temperature response in the most climate sensitive high-latitude regions (polar amplification). This outcome not only contributes to the long-lasting discussion about the potential for polar ice during greenhouse conditions but also offers a strategy to investigate other periods with comparable temperature-$pCO_2$ boundary conditions and inconsistent proxy reconstructions. The strength of the presented best-fit approach between modelling and multi-proxy data compilation is that it is largely decoupled from uncertainties and limitations of individual proxy estimates and other constraints (chronology, paleobathymetry and paleolatitude). Instead, the approach statistically identifies one solution where all available data and model configurations converge towards a single (most) plausible scenario. The described strategy implies that the best-solution approach can be adapted as new evidence and understanding emerges, without being skewed or biased by individual observations. The ongoing discussion on SST estimates from GDGT-based $TEX_{86}$ paleothermometry is one area where a consensus cannot easily be reached, leading to inconsistent and often opposing interpretations. We propose that moving away from such individual observations and discussions towards more integrated solutions, as shown in this study, bear large potential to better recognize large scale inter-relationships and quantify climate and environmental change back in time, setting further constraints for projecting the future.

## Methods

**Calcareous nannofossils and benthic foraminifera**. A total of 69 samples were investigated for calcareous nannofossils; samples were prepared using the simple smear slide technique and examined under polarising light microscope, at ×1250 magnification. Notably, we adopt a revised[56] timescale[57] for the position of the Valanginian/Hauterivian boundary relative to polarity chrons (Supplementary calcareous nannofossils section and Supplementary Fig. 1). Samples for benthic foraminifera analysis were disaggregated in hydrogen peroxide ($H_2O_2$), washed under tap water over >63 μm mesh and dry sieved. All residues of the size-fraction >63 μm were picked, identified, counted, and permanently stored in Plummer slides for benthic foraminifera (Supplementary 'Benthic foraminifera' section).

**Carbon isotope and TOC analysis**. Bulk dried and ground samples ($N = 81$, avg. resolution ~25–50 kyr) were analyzed for bulk organic stable carbon isotope ratios ($\delta^{13}C_{org}$) and TOC at Iso-Analytical, Crewe Cheshire UK, by a Europa Scientific Elemental Analyser–Isotope Ratio Mass Spectrometry (EA–IRMS). Weighed powdered samples were acidified with 2 M hydrochloric acid, mixed, left for 24 h, then washed and oven dried at 60 °C. Weighed samples were loaded into an autosampler. The temperature of the furnace was held at 1000 °C, reaching up ~1700 °C in the region of the sample. Repeat analysis on check samples (e.g., IA-R005 and IA-R006) and the reference material used during $\delta^{13}C$ analysis (e.g., IA-R001-Iso-Analytical standard wheat flour), ensure standard deviation <0.1. All reference materials were calibrated against and traceable to an inter-laboratory comparison standard distributed by the International Atomic Energy Agency (IAEA), Vienna (e.g., IAEA-CH-6). Carbon isotope results are shown in the Vienna Pee Dee Bee δ notation (‰ VPDB) and TOC results are reported in weight percentage (%) in the Supplementary Data Appendix.

**Glycerol dialkyl glycerol tetraethers extraction and $TEX_{86}$ analysis**. Extraction of freeze-dried powdered samples ($N = 48$; 1–5 grams) was performed by Dionex accelerated solvent extraction (DIONEX ASE 350) using a mixture of dichloromethane (DCM)/methanol (MeOH) (5:1, v/v) at a temperature of 100 °C and a pressure of 69 ± 10 psi. Sulfur-free extracts (desulfurization by acid-activated copper turnings) were purified by column chromatography over self-packed silica gel (deactivated with 1% ultrapure $H_2O$) columns using hexane, hexane:DCM (2:1,

v/v), and MeOH as subsequent eluents. The MeOH-fraction containing the GDGTs was dissolved in hexane/isopropanol (95:5, v/v) and 20 μL of $C_{46}$ GDGT tetraether idol lipid standard was added prior to filtering through 0.45 mm Teflon filters. Samples were analysed at the University of Cologne using an Agilent 1290 Infinity ultra high permormance liquid chromatography (HPLC) paired with an Agilent GDGT 6460 Triple Quadrupole MS system. Published chromatographic conditions were applied[58] in selected ion monitoring mode, following previous work for GDGTs identification[23,26]. Peak areas were integrated and calibrated against the internal $C_{46}$ standard. The $TEX_{86}$ index (Eq. 1) was calculated as originally defined[23]:

$$TEX_{86} = \frac{[GDGT-2] + [GDGT-3] + [Cren']}{[GDGT-1] + [GDGT-2] + [GDGT-3] + [Cren']} \quad (1)$$

We note that our high-resolution record from Site 692 is consistent with 5 available published sample data for the study stratigraphic interval from same site location[17]. Repeat analysis of an in-house standard and the application of published analytical techniques[23,26] suggest that the analytical error associated with $TEX_{86}$ analysis is well below the observed variable $TEX_{86}$ range (0.77–0.63) and the mean $TEX_{86}$ decrease from the Weissert warmest interval (~0.74 between A and B in Fig. 2) to the Weissert cold end (~0.66 in C in Fig. 2) documented at Site 692. Therefore, the highest error uncertainty in SST estimates comes calibration errors themselves (see also ref. [27]).

**$TEX_{86}$ calibration.** Based on a published compilation of Cretaceous $TEX_{86}$ data and calibration methods[27], we present in our Supplementary Data Appendix all temperature results based on different calibrations for the Valanginian sites (603[17], 534[17], 766;[17] and 692[17] and this study), including the BAYSPAR calibration model that reports in the main text the upper end $TEX_{86}$ estimates max-$TEX_{86}$ calculated in our analysis. Notably, following a recent study[24] it was further applied a regional calibration, based on a deep-water $TEX^{H}_{86}$ dataset for deep restricted modern basins[42] (Eq. 2), to the restricted Sites 534 and 603, and 692.

$$TEX^{H}_{86} - \text{Deep restricted basin SST } (^{\circ}C) = 56.3 \times (TEX^{H}_{86}) + 30.2; \pm 1 ^{\circ}C \quad (2)$$

**Glycerol dialkyl glycerol tetraethers screening.** A strict GDGT screening procedure was applied (see ref. [27]) to rule out secondary effects, such as thermal alteration, overprint by terrestrial input, and methanogenesis. To this end, we verified that the following indices are within the boundaries deemed to represent suitable organic matter for $TEX_{86}$-based SST reconstructions: a Methane Index (MI) < 0.5, GDGT-0 percentage >67% and the crenarchaeol index ($f_{Cren':Cren'}$ + Cren) < 0.25. The laminated sediments and high TOC values (3.2–12.5%) in the analysed samples indicate deoxygenated bottom waters in the restricted Weddell Sea basin. To test for possible physiological controls on $TEX_{86}$ we calculated the Ring Index (ΔRI)[59] during our initial quality screening. This measure is explicitly designed to identify whether $TEX_{86}$ samples might be significantly influenced by non-temperature factors, like low dissolved oxygen levels. Importantly all $TEX_{86}$ samples show |ΔRI| values well below the threshold of 0.3 (maximum of 0.16) and therefore indicate an overall dominant control of environmental temperature on measured $TEX_{86}$. Note that the branched and isoprenoid tetraether (BIT) index was not calculated in our samples because the concentrations of branched GDGTs were below detection limit. We thus assume the BIT index to be 0 or very low for all investigated samples as shown in previous work at this site[17]. Further details in Supplementary 'Organic matter quality and thermal maturation' section.

**Climate model simulations.** Climate model simulations are based on published work[46] and were performed with the coupled atmosphere-ocean-vegetation climate model HadCM3BL-M2.1aD[60]. The model uses a horizontal resolution of 3.75 ° longitude by 2.5 ° latitude with 19 vertical levels in the atmosphere and 20 ocean depth levels. Two simulations employing a Valanginian (~138 Myr) paleogeography[61] at ×2 and ×4 pre-industrial atmospheric $CO_2$ concentrations (e.g., 560 and 1120 ppm) build the foundation for the model-data comparison. Atmospheric $CO_2$ levels during the Valanginian–Hauterivian are currently not well constrained but were probably lower than during the peak greenhouse conditions of the mid-Cretaceous[10,50], and most likely between ×2 to ×4 preindustrial level[10,50,51]. The general circulation model is coupled to the dynamic global vegetation model TRIFFID via the land surface scheme MOSES 2.1. Orbital parameters are set to present-day values, but with a reduced solar constant of 1349.1 W/m² following published work[62].

The ×4 simulation is initialized with a globally homogeneous salinity of 35 ppt and an idealized zonal mean ocean temperature profile that changes with latitude[61]. The ×2 simulation is branched off at model year 422 and both integrations are continued for a further 10,000 years. Remaining model temperature drifts in both simulations are very small with radiation imbalances at the top of the atmosphere over the last 100 years <0.05 W/m² and changes in volume-integrated ocean temperatures over the last 1000 years <0.04 °C. We note that both simulations show a continuous increase in global mean ocean salinity, even though the influence of this drift on the resulting ocean circulation and surface temperature distribution is probably small (see discussion in ref. [46]).

We follow published methods[46,63] to calculate global mean surface temperature (GMST) estimates for each site by comparing the reconstructed Valanginian ocean temperatures with the simulated sea surface temperatures (SSTs) at the individual paleopositions (Supplementary 'Global mean surface temperature and pCO2 estimates' section and Supplementary Tables 1 and 2).

## Data availability

The $TEX_{86}$ and stable carbon isotopes data and the complete calcareous nannofossil range chart generated in this study are provided in the Supplementary Source data file. Source data are provided with this paper.

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

## Acknowledgements
We thank the ODP repository teams in Bremen and Kochi for providing sample material for this project. We also thank Eni Spa for financial support and permission to publish this study. The project has strongly benefitted from technical support by M. Pegoraro in Milan and D. Warok in Cologne. Funding was provided by Erasmus+ Traineeship scholarship to LC, the Milano group benefited of the fund PRIN 2017RX9XXXY awarded to E.E. and Eni SpA Research & Development provided financial support for research to E.E. and T.W. The authors also acknowledge the financial support from the University of Milan through the APC initiative and from the Italian Ministry of University (MUR) through the project "Diparti-menti di Eccellenza 2018–2022, Le Geoscienze per la Società: Risorse e loro evoluzione".

## Author contributions
L.C. conducted all isotopic and molecular geochemical analyses and led the writing of the study. T.W. conceptualized the study, established the data-model synergies, and co-led the writing, in close partnership with E.E. C.B., G.G. and V.G.-G. conducted chronos-tratigraphic and paleoenvironmental analyses, led by E.E. D.L. and A.F. provided all climate model simulations, S.S. performed the model-data comparison and led the writing of the modelling part. S.F. contributed to the model-data comparison. The biomarker work and TEX86 analyses conducted by L.C. were supported by W.D., J.R., P.H. and O.E. S.T. contributed to the revision and discussion of palynology data available at ODP site 692. All co-authors contributed to the writing of the manuscript.

## Competing interests
The authors declare no competing interests.
