## [Peer Review File · Nature Communications]

REVIEWER COMMENTS

Reviewer #1 (Remarks to the Author):

The m/s explores the Valanginian Weissert event, via new TEX86 data and a compilation of other proxy data and observations. The m/s as is stands is well written although there are a number of points that require some revision. These suggested amendments are as follows:

The title refers to an Antarctic perspective - is this really the case, the data come from ~55 degrees south and much of the proxy data is from tropic and northern latitudes.

The new high-resolution data come the Weddell Sea (East Antarctica), I was hoping to see a simplified paleogeographic reconstruction of the Valanginian and other key DSDP/ODP sites sampled in this study and others.

The TEX86 profile from ODP Site 692 is described as documenting the warmest TEX86 values in the Late Berriasian-Early Valanginian interval & 2 cool episodes coinciding with the Weissert CIE end and a second at the Valanginian/Hauterivian boundary. These results/trends are at the limits of the technique, but I don't see discussion of the analytical error in TEX86 data.

Published TEX86 data from ODP Site 766 offshore Australia also documents variability in TEX86H values. Although these data are slight slightly younger than the Weissert CIE, the datasets do overlap. In the Site 766 data, a greater degree of variability is seen, for example a shift to warmer temperatures at the Valanginian - Hauterivian boundary - why is this not seen in the TEX86 profile from ODP Site 692.

To explore a seasonal warm bias, brought about a seasonality linked to available sunlight and food web dynamics etc., which yes has the potential to skew high-latitude proxy reconstructions towards a seasonal rather than annual mean temperature, the study simulated the warmest 3-month mean temperatures also at sites poleward of 45 degrees. Whilst this approach has some merit, at 45-55 degrees, are such proxies truly going to be that seasonally biased (i.e., just to 3 months) given that the light regime at such latitudes is not so compromised as in polar areas with lengthy periods without daylight. Even with just capturing data for simulated summer SSTs, these estimates still show temperatures cooler than the proxy data presented.

The Fig. 2: shows Valanginian multi-proxy ocean temperature reconstruction and cooling evidence across the Weissert Event. Cooling evidence in sub-polar regions is derived from dropstones, glendonites and tillites. This is somewhat misleading as it looks as if these proxies are restricted to the Late Valanginian and Early Hauterivian. In fact, glendonites are quite common in the Early Valanginian. Likewise, A Late Valanginian tillite is implied, whereas the only described example is the Livingston Tillite Member of the Cadna-owie Formation, South Australia. These sediments have been dated by palynology as Berriasian to Valanginian.

Reviewer #2 (Remarks to the Author):

This paper uses the organic paleothermometer TEX86 to reconstruct high-latitude sea-surface temperatures in the Early Cretaceous. The methods and interpretations are sound, and the paleoclimate story is appropriate for Nature Communications. However, I have concerns regarding the proxy calibration.

TEX86-H is an outdated calibration. Citing O'Brien et al. (2017) as claiming that it is the preferred calibration is incorrect as this paper also uses the linear calibration. The BAYSPAR calibration is becoming favored for Cretaceous studies (e.g. Naafs & Pancost, 2016; O'Connor et al., 2019), even though it produces extremely high temperatures. There is no biological relationship supporting the TEX86-H calibration, and just because linear calibrations do not match climate models does not mean that the calibration is wrong.

If the authors want to follow the path of generating a new calibration, then I suggest submitting this manuscript elsewhere, where the focus of the paper is on both the climate reconstruction and the methods. Otherwise, I strongly suggest that the BAYSPAR calibration is applied and the focus be on the climate story.

Reviewer #3 (Remarks to the Author):

Review

Nature Communications manuscript NCOMMS-21-00870-T

Impact of global cooling in the Early Cretaceous high CO₂ world: the Antarctic perspective
by Cavalheiro et al. (2021)

This paper presents a new and exciting view on an Early Cretaceous carbon cycle perturbation - the Valangin Weissert Event - providing evidence that the associated drop in atmospheric pCO₂ might have favoured the build-up of local Antarctic ice, consistent with geologic evidence. To a new TEX86 data set of the early Wedell Sea basin a recently published TEX86 - SST relationship for restricted ocean basins is applied, which offers the possibility of a more realistic temperature reconstruction in such specific environments. Whilst these findings are novel and of interest to a wider community, and therefore potentially publishable in Nature Communications, I think more work is needed to demonstrate that the reconstructed SST are robust and independent of non-thermal effects. I have the following concerns:

1. Although the application of a TEX86-SST relationship from restricted proto-ocean basin instead of the global calibration is highly acknowledged, it is questionable that the calibration from the Eastern Mediterranean (Kim et al. 2015, GCA) used here is valid for the paleoenvironment of the early Cretaceous Weddell Basin. The laminated sediments in the investigated ODP site 692 section along with the high TOC content call for dysoxic or even anoxic conditions in the narrow basin. Pollen studies of site 692 indicate humid climate conditions around the proto-ocean, suggesting a stratified water column like in the Black Sea today instead of an anti-estuarine circulation as in the Mediterranean. In a culture experiment Qin et al. (2015, PNAS) showed that oxygen concentration is at least as important as temperature in controlling TEX86 values, varying as much as 12 °C from the preset incubation temperatures. Although the expected direction of the deviation is similar to that of the eastern Mediterranean calibration (less dissolved O₂ would lead to a higher temperature bias), it is not clear, to what extent the dysoxygenation might have influenced the ODP 692 TEX86 values. Furthermore, the longterm temperature trend during the 700 ky Weissert Event might be related to changes in the dissolved oxygen rather than to SST changes.
2. Since the presented new SST reconstruction of ODP site 692 would be the only reliable Valanginian temperature record in the Southern Hemisphere (given the fact that the ODP 766 TEX86 data by Littler et al. 2015 cover only the late Valanginian interval and exhibit some extraordinary and difficult to explain values close to the Valanginian-Hauterivian boundary), I urge the authors to provide an independent temperature proxy at Site 692 to validate their TEX86 SST reconstruction. Since the presented model experiments simultaneously produce land surface temperatures along with oceanic SST, a palynofloral estimate would help to better pin the zonal mean temperature gradient of the investigated model experiments. Already in early investigations of site 692 sediments it was stated that pollen and spores are well preserved at this site (Mohr 1990, Proc. ODP 113 Sci Res).
3. I agree with the authors that the Valanginian Weissert Event is an excellent period to give insights into relationships between climate and atmospheric CO₂ drawdown. But the 700-ky SST trend of 3.5°C decrease between the warm and the cold end of the event doesn't fit to the δ¹³C_{org} record, if the latter is taken as a proxy for carbon burial during this period. One would expect an immediate drop in SST associated with the early onset of the Weissert CIE (A to B in Fig. 1), instead a slight warming is indicated. It would be helpful to have some geological evidence that indeed an enhanced bioproductivity triggered by volcanic iron fertilization was responsible for the persistent CO₂ drawdown during the Valanginian.

Minor comments:

Line 29. Already in the summary some more information about the paleoposition of the investigated ODP site 692 (latitude, semi-enclosed basin, water depth) should be provided, to demonstrate the unique and important contribution of this single Southern Hemisphere record to the existing Lower Cretaceous SST compilations (e.g. Charbonnier et al. 2020).

Line 30. Also the SST drop indicated by TEX86 should be described more precisely, since the maximum decrease in SST is only reached towards the end of the Weissert CIE.

Line 48. Please shortly summarize, why the published Valangian records are inconsistent in their interpretation.

Line 108. As mentioned above, it should be distinguished between restricted settings in an subtropical proto-North Atlantic (DSDP Sites 534 and 603) with presumably arid climate conditions, and those in the early Cretaceous Weddell Basin (ODP Site 692) with humid conditions in a rather cool temperate climate.

Line 131. Since non-thermal effects have been shown to be of significant influence for TEX86 SST reconstructions, it is not easy to accept that over a 700 ky period the environmental conditions were almost constant. Are there any hints from lithology that this is the case in ODP 692 10R and 12 R sediments?

Line 141. I would prefer to see the original paleotemperature estimates of the different sites discussed in comparison to ODP 692 record, instead just presenting summarized trends.

Line 197. I agree that the highest degree of consistency between model temperatures and the new proxy compilation is achieved by the application of a regional TEX86 calibration for restricted Mesozoic ocean basins at the North Atlantic sites, but still there is a significant offset to warmer temperatures in ODP site 692 temperatures in the warm interval as well as in the cold end of the Weissert Event.

Line 226. If you assume meltwater to have lowered salinity significantly, is there any indication for that in the available oxygen isotope records of the according high latitude basins?

Fig 1. A Valangianian paleomap of the Weddell Sea Area should be included (as an inset) to show the special position of the investigated core in the proto-ocean basin.

Fig. 2. Although I acknowledge the clear layout of the figure, I would prefer to see the original paleotemperature estimates of only those sites, which cover the entire Late Valangianian to early Hauterivian period, and which present continuously the zonal temperature evolution of the southern subpolar, the subtropical and the northern subpolar latitudes. Along with that the map of the used sites (Suppl Fig. 1) should be presented already here in the main document.

Fig S1. Just a typo: At the latitude scale, 30°S is indicated as 30°N

“Impact of global cooling on Early Cretaceous high $p\text{CO}_2$ world during the Weissert Event” - manuscript

We thank the three reviewers for the detailed and constructive comments and suggestions. They have strongly helped to improve the focus and structure of the manuscript while presenting more clearly the core scientific results and outcomes of our research. Furthermore, we have extended the overall length of the text and reorganized individual components following NC guidelines. We are now confident that the revised manuscript addresses in full all critical points raised and we are looking forward to your response.

Before we address the comments point-by-point below, we want to expand on two general points in order to (re-)position our revised manuscript.

Point one: much critique has been focused around the TEX_{86} temperature calibration and its interpretation; we fully share all these concerns and we are well aware of the multitude of limitations and complications this proxy has, for each of the commonly used calibration solution. The ongoing discussion on SST estimates from GDGT-based TEX_{86} paleothermometry is one area where a consensus cannot easily be achieved, leading to inconsistent and often opposing interpretations. Therefore, following the shared reviewers’ suggestions, we now include a new text section “**absolute derived TEX_{86} SSTs** (lines 170-224)”, using the BAYSPAR model as reference (Tierney and Tingley 2014, 2015). We also include the outcomes from all widely applied TEX_{86} calibrations summarised in the Supplementary Data Appendix. We point out that in our analysis the BAYSPAR model provides the upper end estimate of absolute TEX_{86} -derived ocean temperatures (i.e., “max TEX_{86} ” estimates discussed in the article) in all study Valanginian sites. Following a recent study (Steinig et al., 2020), we report in the text also the results of applying a restricted basin calibration (Kim et al., 2015) in the study sites from the restricted proto-North Atlantic and proto-Southern Ocean (ODP Site 692, this study); we therefore show that this calibration yields results as a possible lower end estimate (i.e., “min TEX_{86} ”) of absolute SSTs in these restricted environments.

Point two: we want to emphasise that our study does not question any of these calibrations (i.e., BAYSPAR vs. restricted basin calibration) and we have no *a priori* preference for either of them. We instead argue and demonstrate that the novelty and strength of our new approach is that we combine TEX_{86} SST (with its diverse calibration outcomes) with other available temperature information from proxy or geological data for the Weissert Event and match this integrated ‘data view’ with climate modelling results. The modelling outcomes are not driven by any of the proxy or geological data and as such are independent, however, they are tightly bound by a new and robust chronological context introduced in our study. This integrated approach enables us to identify the ‘best fit’ between climate simulations and one specific data combination. We argue that this single combination represents the most consistent solution of the Weissert temperature distribution, globally, and its evolution across the cooling event. We emphasise that this best-fit approach is statistically robust and independent from any personal view or preference. The outcome, however, indeed favours one TEX_{86} solution. For the Weissert Event, this solution builds on two different TEX_{86} calibrations, one for sites in open ocean conditions (ODP Site 766) and one for sites in restricted ocean basins (DSDP Sites 603 and 534, and ODP Site 692). Therefore, we point

out that the “min TEX₈₆” results in our study site (Site 692, Weddell Sea) and in the North Atlantic sites (Sites 603 and 534) are more consistent to climate model outcomes and independent proxy reconstructions. To our best knowledge, this solution has not been proposed before, but opens new directions of proxy data interpretation. Within this context we present the first, fully validated high-resolution SST record of the Weissert Event for the sub-Antarctic Weddell Sea. As explained in the revised manuscript the absolute temperature drop of around 3-4°C from the Weissert onset to its end is similar for all different calibration methods and as such a robust result that carries weight.

In summary, we are confident that only moving away from such individual observations and discussions towards more integrated solutions allows us to simulate the level and fluctuation in atmospheric *p*CO₂, global cooling and potential for polar ice built-up, all within firm and well defined geological, chronological and climatic boundaries. As shown in this study, this best-fit approach bears large potential to better recognize large scale inter-relationships and quantify climate and environmental change back in time, setting new constraints for projecting the future.

Looking forward, as new or improved temperature data may emerge for the Weissert interval the unique solution presented in this study may prompt further improvement and adjustment; this would be good practise and ensure scientific advancement. We conclude that the concept presented in our study is innovative and advanced, compared to previous published studies. It may even carry elements that could serve as a template for other integrated paleoclimatic and paleoenvironmental research of major carbon-climate perturbations.

Other comments of each reviewer are addressed point-by-point in the following check list:

Reviewer #1 (Remarks to the Author):

- The m/s explores the Valanginian Weissert event, via new TEX₈₆ data and a compilation of other proxy data and observations. The m/s as is stands is well written although there are a number of points that require some revision.

We thank the reviewer for this positive and encouraging statement.

These suggested amendments are as follows:

- The title refers to an Antarctic perspective - is this really the case, the data come from ~55 degrees south and much of the proxy data is from tropic and northern latitudes:

*We recognise this criticism and changed the Title to: **Impact of global cooling on Early Cretaceous high *p*CO₂ world during the Weissert Event***

- The new high-resolution data come the Weddell Sea (East Antarctica), I was hoping to see a simplified paleogeographic reconstruction of the Valanginian and other key DSDP/ODP sites sampled in this study and others:

A palaeographic map showing all site locations discussed in this work was initially present in the Supplementary, as Figure S1. Following the reviewer's suggestion we now present it in the main article as Figure 1.

- The TEX₈₆ profile from ODP Site 692 is described as documenting the warmest TEX₈₆ values in the Late Berriasian - Early Valanginian interval & 2 cool episodes coinciding with the Weissert CIE end and a second at the Valanginian/Hauterivian boundary. These results/trends are at the limits of the technique, but I don't see discussion of the analytical error in TEX₈₆ data.

Repeat analysis of an in-house standard and the application of published analytical techniques (e.g., Schouten et al., 2007, see Method section lines 407-430) suggest that the analytical error associated with TEX₈₆ analysis is well below the observed variable TEX₈₆ range (0.77–0.63) as well as the mean TEX₈₆ decrease from the Weissert warmest interval (~0.74 between A-B in Figure 2) to the Weissert cold end (~0.66 in C in Figure 2). We therefore point out that the highest uncertainty in SST estimates comes calibration errors themselves (e.g., see O'Brien et al., 2017). Based on the comparison with other proxy and geological temperature evidence we are as confident as possible that the observed cooling trend at ODP Site 692 is consistent with global evidence and the high latitude cooling trend reported for ODP site 765 offshore Australia (Charbonnier et al., 2020) using oxygen isotopes.

- Published TEX₈₆ data from ODP Site 766 offshore Australia also documents variability in TEX_{86H} values. Although these data are slightly younger than the Weissert CIE, the datasets do overlap. In the Site 766 data, a greater degree of variability is seen, for example a shift to warmer temperatures at the Valanginian – Hauterivian boundary – why is this not seen in the TEX₈₆ profile from ODP Site 692.

*In the revised section “**Contrasting TEX₈₆-derived temperature across the Weissert Event**” (lines 95-121) we discuss the stratigraphic uncertainties and temperature trends of ODP Site 766 in large detail. We emphasise that the $\delta^{13}\text{C}_{\text{org}}$ profile from ODP Site 766 does not unequivocally identify the Weissert CIE and magnetostratigraphy (Littler et al., 2011) suggests an age which is slightly younger than the end of the Weissert CIE. Therefore, on the basis of a detailed chronostratigraphic revision of these records, we conclude that at ODP Site 766 the correlation of the observed short shift to warmer temperatures to other sites needs be interpreted with caution due to large stratigraphic uncertainties. However, it is noteworthy that in contrast to the stable and high TEX₈₆-based temperatures documented in the proto-North Atlantic sites (Littler et al., 2011), ODP Site 766 documents TEX₈₆ variability in the Late Valanginian-Early Hauterivian, which corroborates the temperature variability at ODP site 692 and a higher temperature response of sub-polar regions in contrast to the lower latitudes. Further chronostratigraphic constraints and high-resolution TEX₈₆ analysis at ODP Site 766 or similar high latitude sites (e.g., ODP Site 765; Charbonnier et al., 2020) are needed for a precise correlation of the observed temperature trends.*

- To explore a seasonal warm bias, brought about a seasonality linked to available sunlight and food web dynamics etc., which yes has the potential to skew high-latitude proxy reconstructions towards a seasonal rather than annual mean temperature, the study

simulated the warmest 3-month mean temperatures also at sites poleward of 45 degrees. Whilst this approach has some merit, at 45-55 degrees, are such proxies truly going to be that seasonally biased (i.e., just to 3 months) given that the light regime at such latitudes is not so compromised as in polar areas with lengthy periods without daylight. Even with just capturing data for simulated summer SSTs, these estimates still show temperatures cooler than the proxy data presented.

We acknowledge that previous work (e.g., Bijl et al., 2009; Super et al., 2018; O'Connor et al., 2019) has brought evidence for a seasonal bias affecting geochemical proxies in polar and sub-polar regions but a number of studies (e.g., Davies et al., 2019; Winter et al., 2021) support a warm seasonal bias affecting also mid-high latitude geochemical proxy temperature estimates. Sediment trap measurements of modern planktonic foraminifera reveal spring to summer flux peaks even in the mid-latitude Southern Ocean (~50 °S) that are retained in the sedimentary record (King and Howard, 2005). Therefore, we explore the possibility of a seasonal warm bias affecting geochemical proxy records at mid-to high latitudes at all sites poleward of 45 degrees (lines 246-257 in main text). We do not interpret this latitude as a fixed threshold, but rather as the latitudinal band at which seasonal production patterns might become increasingly relevant for the sedimentary record with strongest signals in polar latitudes. Due to the still limited understanding of these seasonal effects, we apply a rather simple assumption of a similar summer SST bias for all mid to high latitude sites. While the seasonality might be overestimated at the mid latitudes, it might be equally underestimated at the very high latitudes. However, we want to stress that this study does not aim to discuss the ecology assumptions behind different proxy techniques. We rather present a multi-proxy dataset and a resulting best-fit approach between all data sets and climate model outcomes investigating a potential seasonal bias in parallel with different calibration approaches in order to generate a plausible range of absolute sea surface temperatures for each site.

- The Fig. 2: shows Valanginian multi-proxy ocean temperature reconstruction and cooling evidence across the Weissert Event. Cooling evidence in sub-polar regions is derived from dropstones, glendonites and tillites. This is somewhat misleading as it looks as if these proxies are restricted to the Late Valanginian and Early Hauterivian. In fact, glendonites are quite common in the Early Valanginian. Likewise, A Late Valanginian tillite is implied, whereas the only described example is the Livingston Tillite Member of the Cadna-owie Formation, South Australia. These sediments have been dated by palynology as Berriasian to Valanginian.

*We acknowledge that careful stratigraphic re-analysis of available geological records suggests that the supporting glacial evidence from sub-polar regions (Rogov et al., 2017; Vickers et al., 2019; Alley et al., 2019) has to be used with caution due to large stratigraphic uncertainties across the Berriasian to the Early Valanginian (see discussion in lines 160-168 and revised Fig. 3 in main text). We point out (“**High-latitude cooling and ice formation during the end of the Weissert Event**”, lines 279 - 306) that our model supports the possibility for seasonal sea ice formation in the Arctic and around Antarctica for both Weissert pCO₂ scenarios, however to different extents. Our results do not comment on the possibility of glacial deposits preceding the Weissert cold end. We note, however, that they are in agreement with a wide consensus for cooler conditions and limited glaciation in the sub-polar Polar Regions across the Early Cretaceous (e.g., Rogov et al., 2017; Vickers et al.,*

2019; Alley et al., 2019). We explain and bring in multi-proxy and modelling evidence supporting that seasonal sea ice coverage expands during the cool end of the Weisert Event, with largest increases around the Arctic. In Figure 5c we now show that the latitudes of maximum simulated cooling and sea ice extension ($\sim 50 - 70^\circ$ in both hemispheres) coincide with the location of latest Valanginian dropstones, glendonites and tillites (Charbonnier et al., 2020 compilation and paleo-geographic distribution in Figure 1), in combination indicating cooler surface waters, sea ice and drifting ice conditions in the latest Valanginian. This is as far as we are comfortable to push the interpretation of our results. This leaves inevitably uncertainties before and after the Weisert event itself (which we analyse unprecedented large detail) we cannot confidently address in our study.

Reviewer #2 (Remarks to the Author):

- This paper uses the organic paleothermometer TEX₈₆ to reconstruct high-latitude sea-surface temperatures in the Early Cretaceous. The methods and interpretations are sound, and the paleoclimate story is appropriate for Nature Communications. However, I have concerns regarding the proxy calibration.
- TEX₈₆-H is an outdated calibration. Citing O'Brien et al. (2017) as claiming that it is the preferred calibration is incorrect as this paper also uses the linear calibration. The BAYSPAR calibration is becoming favored for Cretaceous studies (e.g. Naafs & Pancost, 2016; O'Connor et al., 2019), even though it produces extremely high temperatures. There is no biological relationship supporting the TEX₈₆-H calibration, and just because linear calibrations do not fit climate models does not mean that the calibration is wrong. If the authors want to follow the path of generating a new calibration, then I suggest submitting this manuscript elsewhere, where the focus of the paper is on both the climate reconstruction and the methods. Otherwise, I strongly suggest that the BAYSPAR calibration is applied and the focus be on the climate story.

We thank you for your comment and following your suggestion we have revised our manuscript using the BAYSPAR model as reference TEX₈₆ calibration, see detailed discussion above introducing our point-by-point response to all reviewers.

Reviewer #3 (Remarks to the Author):

Review

Nature Communications manuscript NCOMMS-21-00870-T

Impact of global cooling in the Early Cretaceous high CO₂ world: the Antarctic perspective by Cavalheiro et al. (2021)

This paper presents a new and exciting view on an Early Cretaceous carbon cycle perturbation - the Valanginian Weissert Event - providing evidence that the associated drop in atmospheric pCO₂ might have favoured the build-up of local Antarctic ice, consistent with geologic evidence. To a new TEX86 data set of the early Weddell Sea basin a recently published TEX86 - SST relationship for restricted ocean basins is applied, which offers the possibility of a more realistic temperature reconstruction in such specific environments. Whilst these findings are novel and of interest to a wider community, and therefore potentially publishable in Nature Communications, I think more work is needed to demonstrate that the reconstructed SST are robust and independent of non-thermal effects. I have the following concerns:

1. Although the application of a TEX86-SST relationship from restricted proto-ocean basin instead of the global calibration is highly acknowledged, it is questionable that the calibration from the Eastern Mediterranean (Kim et al. 2015, GCA) used here is valid for the paleoenvironment of the early Cretaceous Weddell Basin. The laminated sediments in the investigated ODP site 692 section along with the high TOC content call for dysoxic or even anoxic conditions in the narrow basin. Pollen studies of site 692 indicate humid climate conditions around the proto-ocean, suggesting a stratified water column like in the Black Sea today instead of an anti-estuarine circulation as in the Mediterranean. In a culture experiment Qin et al. (2015, PNAS) showed that oxygen concentration is at least as important as temperature in controlling TEX86 values, varying as much as 12 °C from the preset incubation temperatures. Although the expected direction of the deviation is similar to that of the eastern Mediterranean calibration (less dissolved O₂ would lead to a higher temperature bias), it is not clear, to what extent the dysoxygenation might have influenced the ODP 692 TEX86 values. Furthermore, the longterm temperature trend during the 700 ky Weissert Event might be related to changes in the dissolved oxygen rather than to SST changes.

It is difficult if not impossible to disentangle and separate restricted ocean circulation from water column deoxygenation. There is evidence for both in the Cretaceous Weddell Sea, as there is for other young basins, including the Cretaceous proto-North Atlantic for which Mediterranean-like isoprenoid GDGT distributions have been described (Steinig et al., 2020). We concur that the paleoceanographic setting and resulting circulation of the Valanginian Weddell Sea are different from the present-day Eastern Mediterranean Sea. While uncertainties in the paleogeographic boundary conditions and relatively coarse model resolution prevent an adequate simulation of the circulation in the small Weddell Basin, the paleolatitude of ~54 °S is consistent with an overall more humid climate based on modern day climate zonation. While this is clearly different to the modern evaporative Mediterranean Sea, we argue that the increased surface freshwater input combined with deoxygenated

bottom waters is a better analogue to times of sapropel formation in the Mediterranean Sea (see discussion in main text, lines 191-213; further support in Supplementary lines 152-181). Importantly, a downcore study (Polik et al., 2018) showed that regional TEX₈₆-SSTs are consistently warmer (up to 15 °C; their Figure 4) than U^K₃₇-derived temperatures, both inside and outside of Pleistocene sapropels. This is clear evidence that the occurrence of endemic Thaumarchaeota populations in the restricted basin can change the local TEX₈₆-SST relation independent of large swings in water column stratification and oxygenation. We therefore argue that the young and restricted basins of the Early Cretaceous, in analogue to the modern Mediterranean and Red Sea, also provide special environmental conditions enabling specific Thaumarchaeota community structures and potentially different TEX₈₆ export dynamics than commonly considered in the present-day core-top data. The Cretaceous greenhouse climate might have allowed the formation of warm, isolated water masses with temperatures comparable to the modern Mediterranean Sea even in mid latitudes, especially in basins disconnected from the global overturning circulation (e.g. Fig. 4c in Steinig et al. 2020). Restriction therefore seems a common feature for all basins under consideration, modern and past, but the role of ocean deoxygenation and water column stratification in modulating these special environmental conditions is less clear. We do not argue that bottom water anoxia have no influence on TEX₈₆-derived temperatures, but rather that the special Thaumarchaeota community structure in these restricted environments defines a background TEX₈₆-SST relation that is different from the global core-top calibrations.

To our knowledge there is currently no evidence that confirms the culture experiment results from Qin et al. (2015, PNAS) in sedimentary records. Still, to test for possible physiological factors we calculated the Ring Index (Zhang et al., 2016) during our initial quality screening (see Methods, lines 442-456). This measure is explicitly designed to identify whether TEX₈₆ samples might be significantly influenced by non-temperature factors, like low dissolved oxygen levels. Importantly all new TEX₈₆ samples show |ΔRI| values well below the threshold of 0.3 (maximum of 0.16) and therefore indicate an overall dominant control of environmental temperature on measured TEX₈₆. Even in the case of a small additional influence of low dissolved oxygen levels on the Weddell Sea TEX₈₆, reconstructed SSTs would be similarly overestimated as in the present-day Mediterranean Sea (as noted by the reviewer). We conclude that this only further justifies our approach to also generate a new minimum temperature estimate for the Weddell Sea record. The comparison with independent proxy methods and modelling results allows for discussion of the climatic implications of these lower temperatures irrespective of the exact ecological reasoning.

2. Since the presented new SST reconstruction of ODP site 692 would be the only reliable Valanginian temperature record in the Southern Hemisphere (given the fact that the ODP 766 TEX₈₆ data by Littler et al. 2015 cover only the late Valanginian interval and exhibit some extraordinary and difficult to explain values close to the Valanginian-Hauterivian boundary), I urge the authors to provide an independent temperature proxy at Site 692 to validate their TEX₈₆ SST reconstruction. Since the presented model experiments simultaneously produce land surface temperatures along with oceanic SST, a palynofloral estimate would help to better pin the zonal mean temperature gradient of the investigated model experiments. Already in early investigations of site 692 sediments it was stated that pollen and spores are well preserved at this site (Mohr 1990, Proc. ODP 113 Sci Res).

In the revised manuscript we comment on available palynoflora data at ODP site 692 (Mohr 1990); see revised text in lines 147-157. Careful inspection of Mohr's (1990) paper (note that we now include a highly experienced palynologist as co-author of the manuscript) indicates that these palynological data are not diagnostic of the temperature fluctuations reconstructed with TEX₈₆. In fact, palynological data cannot provide quantitative estimates of paleo-temperature changes but rather define the climatic boundary conditions on land. The floral data from ODP 962 suggest the presence of a cool temperate forest on the Antarctic continent, with high moisture levels and strong seasonality with temperature below freezing. The stratigraphic resolution of the data is insufficient to capture the internal structure of the Weissert perturbation, however, we note a palynological assemblage with reconstructed preference for cooler and more humid conditions for the interval characterized by the TEX₈₆ minimum (i.e. coolest temperature), consistent with the Weissert cold end (C in Figure 2). Palynology therefore tends to support our climate interpretation derived from TEX₈₆ SST and climate modelling.

3. I agree with the authors that the Valanginian Weissert Event is an excellent period to give insights into relationships between climate and atmospheric CO₂ drawdown. But the 700-ky SST trend of 3.5°C decrease between the warm and the cold end of the event doesn't fit to the δ¹³C_{org} record, if the latter is taken as a proxy for carbon burial during this period. One would expect an immediate drop in SST associated with the early onset of the Weissert CIE (A to B in Fig. 1), instead a slight warming is indicated. It would be helpful to have some geological evidence that indeed an enhanced bioproductivity triggered by volcanic iron fertilization was responsible for the persistent CO₂ drawdown during the Valanginian.

We are fully aware of the ongoing discussion on the causes that lead to global cooling and persistent pCO₂ drawdown in the latest Valanginian (e.g., Charbonnier et al. 2020; Price et al., 2020; Mutterlose and Wise 2008; Erba et al., 2004). As summarized in the revised introduction (lines 40-44) the Weissert CIE, like other Cretaceous climate perturbations (e.g., McAnena et al., 2013 etc.), is thought to have resulted from extended volcanism (in our case study the Paraná-Etendeka Large Igneous Province; Erba et al., 2004). A carbonate crises in pelagic and neritic environments, local to regional enhanced productivity and oceanic dysoxia/anoxia has been proposed as key drivers to enhanced marine organic carbon (C_{org}) burial, presumably triggering atmospheric pCO₂ drawdown and global cooling (e.g., Weissert and Erba 2004; Erba et al., 2004; Erba and Tremolada 2004; Charbonnier et al., 2020; Price et al., 2020). However, we point out that a massive marine Corg burial episode associated with the Weissert Event has not been documented in previous studies. The record from ODP 692 is the first to provide evidence for very high marine organic carbon burial, however not limited to the period of cooling of the Weissert perturbation. We decided in our study not to speculate about the quantity of enhanced carbon burial in the proto Weddell Sea or the early Southern Ocean, and its potential role in causing global cooling and deflecting carbon isotopes. As a reference point for this aspect, we refer to basin scale estimates of carbon burial in the early Cretaceous (Aptian-Albian) ocean presented by McAnena et al., 2013. These estimates do not specifically target the Weissert/ Valanginian-Hauterivian boundary but emphasise the role of young and restricted ocean basins in the wake of the breakup of Pangaea as primary carbon sinks, with the early South Atlantic and Southern Ocean standing out. We want to underline here again that the aim of our study is neither to discuss the triggering factors of the Weissert Carbon Isotope Excursion nor the driving mechanisms behind the Late Valanginian global cooling and pCO₂ drawdown. Instead, we

present a novel approach to estimate of the global mean surface temperatures and associated pCO₂ concentrations for the different phases of the Weissert Event based on multi-proxy and modelling evidence.

Minor comments:

Line 29. Already in the summary some more information about the paleoposition of the investigated ODP site 692 (latitude, semi-enclosed basin, water depth) should be provided, to demonstrate the unique and important contribution of this single Southern Hemisphere record to the existing Lower Cretaceous SST compilations (e.g. Charbonnier et al. 2020).

We have slightly modified the abstract, including your suggestions in lines 22-23, e.g., “...the semi-enclosed shelf Weddell basin, offshore Antarctica (paleo-latitude ~54 °S and paleo-water depth ~500 meters)...”

Line 30. Also the SST drop indicated by TEX₈₆ should be described more precisely, since the maximum decrease in SST is only reached towards the end of the Weissert CIE.

We have slightly modified the abstract, including your suggestions in lines 23-24, e.g., “The TEX₈₆ data document a ~3–4°C drop in SST coinciding with the Weissert cold end”.

Line 48. Please shortly summarize, why the published Valanginian records are inconsistent in their interpretation.

*We have modified the revised text as follows (Introduction, lines 47-50): “many temperature proxy records are inconsistent in their interpretation (e.g., contrasting ocean temperature reconstructions based on TEX₈₆, oxygen isotopes and Mg/Ca measurements) not systematically compared with independent evidence, such as climate model simulations, and not aligned against one unified chronological framework”. Further discussion in results section “**Integrating global ocean temperatures and cooling evidence**”.*

Line 108. As mentioned above, it should be distinguished between restricted settings in a subtropical proto-North Atlantic (DSDP Sites 534 and 603) with presumably arid climate conditions, and those in the early Cretaceous Weddell Basin (ODP Site 692) with humid conditions in a rather cool temperate climate.

We have included your suggestion in main text in lines 199-204, expanding also the discussion and comparison with Pleistocene sapropels from the Mediterranean Sea (Polik et al., 2018) (further discussion in Supplementary lines 164-181).

Line 131. Since non-thermal effects have been shown to be of significant influence for TEX₈₆ SST reconstructions, it is not easy to accept that over a 700 ky period the environmental conditions were almost constant. Are there any hints from lithology that this is the case in ODP 692 10R and 12 R sediments?

In the original report (Barker and Kennet 1988) lithology is described in detail but no relevant changes are reported in the overall Valanginian – Hauterivian calcium carbonate-rich and finely laminated black shale section recovered at ODP Hole 692B.

Line 141. I would prefer to see the original paleotemperature estimates of the different sites discussed in comparison to ODP 692 record, instead just presenting summarized trends.

We have slightly modified the structure of the article as explained above. Therefore, original paleotemperature estimates of the different sites presented in comparison to ODP 692 record are discussed in the sub-paragraph “Integrating global ocean temperatures and cooling evidence” (lines 123-168). Furthermore, a summary table of all available temperature records encompassing the Weissert CIE (Figure 3) is reported in Supplementary Table 1 (S1).

Line 197. I agree that the highest degree of consistency between model temperatures and the new proxy compilation is achieved by the application of a regional TEX₈₆ calibration for restricted Mesozoic ocean basins at the North Atlantic sites, but still there is a significant offset to warmer temperatures in ODP site 692 temperatures in the warm interval as well as in the cold end of the Weissert Event.

We agree that there remains a significant offset (~ 6-8 °C) between reconstructed and simulated annual mean temperatures at ODP Site 692. But we also show that the difference significantly narrows (<2 °C) once we compare the reconstructed temperatures with 3-month summer means from the model (light red squares in new Fig. 5). We therefore conclude, as stated in the main text, that we find the best model-data fit for a regional TEX₈₆ calibration in combination with a seasonal warm bias in high-latitude sites. As explained above in our rebuttal, our approach is unbiased by any specific proxy or calibration but aims to statistically define a unique configuration where all available data result in a best-fit solution.

Line 226. If you assume meltwater to have lowered salinity significantly, is there any indication for that in the available oxygen isotope records of the according high latitude basins?

The oxygen isotope records should register a salinity decrease. Unfortunately, there are no such data available for high latitude (sub-Antarctic) basins and - as pointed out in a recent overview (Charbonnier et al., 2020) - there are significant discrepancies in recent Early Cretaceous global sea level changes and variations in ocean water salinity recorded in high-latitude basins from oxygen isotopes (lines 302-306 main text). Instead, we now refer to paleoecological data of braarudosphaerids that support a freshening of surface waters during the Weissert Event and in the warmer interval following the cold phase across the Valanginian/Hauterivian boundary, recorded at ODP 692 (new data from our study), ODP sites 766 and 765 and across the high latitudes in the northern hemisphere.

Fig 1. A Valanginian paleomap of the Weddell Sea Area should be included (as an inset) to show the special position of the investigated core in the proto-ocean basin.

A palaeographic map showing all site locations discussed in this work was present in the Supplementary, as Figure S1. Although, following rev#1 suggestion, we have thought to present it in the main article as Figure 1.

Fig. 2. Although I acknowledge the clear layout of the figure, I would prefer to see the original paleotemperature estimates of only those sites, which cover the entire Late Valanginian to early Hauterivian period, and which present continuously the zonal temperature evolution of the southern subpolar, the subtropical and the northern subpolar latitudes. Along with that the map of the used sites (Suppl. Fig. 1) should be presented already here in the main document.

We thank the reviewer for the suggestion, however, we think that the aim of Figure 2 (now Figure 3 in main text) is to present a wide picture of the available Valanginian-Early Hauterivian temperature records that we have positioned in detailed in our revised chronostratigraphic framework of the Weissert Event. A summary table of all available temperature records encompassing the Weissert CIE is reported in Supplementary Table 1 (S1), reporting average temperature values for the Weissert warm interval (A-B in Figure 2) and the Weissert cold end (C in Figure 1). We also remark that most of the quoted temperature records are shown in detail in the review paper of Charbonnier et al., 2020.

Fig S1. Just a typo: At the latitude scale, 30°S is indicated as 30°N:

Thank you for pointing it out. The palaeographic map showing all site locations is now corrected in the main article as Figure 1.

REVIEWER COMMENTS

Reviewer #1 (Remarks to the Author):

the authors of the revised manuscript have clearly responded to the comments provided by reviewers. The suggestions I made have been fully addressed in the revision.

Gregory Price

Reviewer #2 (Remarks to the Author):

Reviewing this manuscript for the second time, I'm pleased with the improvements the authors have made. There is less focus on the proxy calibration and more on the climate story and potential proxy-model discrepancies. The authors have addressed my concerns from the previous review. As such, I recommend minor revisions, though these are mostly for clarity.

Minor points:

- Line 27: 'that' should be 'which'
- Line 28: the authors state that CO₂ drops over a period of 700kyr. This statement implies that the CO₂ then stays at this new level, but the $\delta^{13}\text{C}$ curve suggests further changes in the carbon cycle. Does CO₂ return to the previous level after the event?
- Line 102: why are the intervals separated by a warming interlude?
- Line 129: I'm not sure 'consistently' is the right word; perhaps 'consistent'?
- Line 131: 'Southern Arctic and Boreal' Not sure what this refers to? I've never heard the first term
- Line 151: I'm not sure 'enumerates' is the right word; perhaps 'comprises'?
- Line 154: should be 'worth noting'
- Line 192: Should be 'GDGT'
- Line 221: the term 'CIE end' is confusing. Do you mean the recovery from the CIE? Consider rewording
- Line 302 (and check throughout): don't begin a sentence with this/it/they/etc. without clarifying what you are referring to
- High-latitude restricted basins are often cooler than the latitudinal mean (Judd et al., 2020, GRL). This finding might help support your argument that the TEX₈₆-SST relationship is problematic in these settings
- Restriction does not necessarily mean stratification. Do you have any geochemical or lithological data to support stratification of the water column?
- The depth of GDGT production may be assessed using the [2]/[3] ratio (Taylor et al., 2013, GPC). I suggest you investigate this geochemically if you are to make the claim that it affects your temperature reconstructions.

Reviewer #3 (Remarks to the Author):

Review of revised manuscript (June 2021)

Nature Communications manuscript NCOMMS-21-00870-A
by Cavalheiro et al. (2021)

I am pleased to see the authors addressed my comments so comprehensively. Although the the TEX₈₆ temperature calibration and its interpretation is still problematic, the according discussion is significantly improved. However, I am not confident with the reply that palynological data cannot provide quantitative estimates of paleotemperature. To my knowledge, the NLR approach allows to reconstruct mean annual as well as warmest month mean temperatures. Therefore I suggest to investigate at least a few samples of ODP 692 in respect to produce independent paleotemperature data for the Weissert Event.

Just a minor mistake:

Line 616. The name of the second author of ref 46 is Elling, not Felix.

REVIEWER COMMENTS

Reviewer #1 (Remarks to the Author):

the authors of the revised manuscript have clearly responded to the comments provided by reviewers. The suggestions I made have been fully addressed in the revision.

Gregory Price

We thank the reviewer for this very positive and encouraging conclusion.

Reviewer #2 (Remarks to the Author):

Reviewing this manuscript for the second time, I'm pleased with the improvements the authors have made. There is less focus on the proxy calibration and more on the climate story and potential proxy-model discrepancies. The authors have addressed my concerns from the previous review. As such, I recommend minor revisions, though these are mostly for clarity.

Minor points:

- Line 27: 'that' should be 'which'

Thank you for pointing out the mistake, now corrected.

- Line 28: the authors state that CO₂ drops over a period of 700kyr. This statement implies that the CO₂ then stays at this new level, but the $\delta^{13}\text{C}$ curve suggests further changes in the carbon cycle. Does CO₂ return to the previous level after the event?

The focus of this study is the observed Late Valanginian global cooling and associated pCO₂ drop over a time interval of about 700 kyrs. We do not elaborate on temperature trends before or after this cooling period. Following the long-term cooling trend targeted in our study, temperatures will have generally warmed, however with swings on a longer term temperature trend. There are good reasons why we do not discuss these temperature undulations following the cooling. We recognize a brief warming interlude in different sections (ODP Site 692, ODP site 766 and in other Tethyan sections), however the resolution and stratigraphy of the analysed records impede reliable correlations of such relatively short and low-amplitude SST fluctuations. We therefore do not feel in the position to disentangle the local or global nature of such warming interludes, assess possible associations with pCO₂, or speculate about their causes (e.g. orbital forcing). Based on the available literature and our new data from ODP Site 692 there is no evidence that SST reached pre-Weissert levels closely following the maximum cooling or following the warming interlude. We consequently speculate that pCO₂ levels also did not reach pre-Weissert conditions in the Early Hauterivian. This conclusion is not fully satisfactory, as we are fully aware, however, we decided to only climate variations that we can confidently correlated, both stratigraphically and spatially.

- Line 102: why are the intervals separated by a warming interlude?

See discussion above.

- Line 129: I'm not sure 'consistently' is the right word; perhaps 'consistent'

Changed to "consistent".

- Line 131: 'Southern Arctic and Boreal' Not sure what this refers to? I've never heard the first term

Corrected to “in the in the southern and arctic part of the Boreal Realm (~38–65 °N) (Meissner et al., 2015).

- Line 151: I’m not sure ‘enumerates’ is the right word; perhaps ‘comprises’

Changed to “comprises”.

- Line 154: should be ‘worth noting’

Corrected to “worth noting”.

- Line 192: Should be ‘GDGT’

Corrected to “GDGT”.

- Line 221: the term ‘CIE end’ is confusing. Do you mean the recovery from the CIE? Consider rewording

*We are sorry for any confusion caused and we are aware of slightly different definitions of CIEs and their main turning points in the literature. As stated in the manuscript, for the Weissert CIE we follow the definition of Erba et al., 2004: “**the beginning and the end of the event** are placed at the base of the positive excursion (uppermost part of magnetic chron CM12) and at its climax (upper part of magnetic chron CM11), respectively”. Therefore, following this definition, we define “the onset of the Weissert CIE” with the reference point A in Figure 2 and point C as the “end of the Weissert CIE”. The ‘recovery period’, marking the return of isotope trends towards pre-perturbation levels, follow point C, the end of the Weissert perturbation. We have now clarified that in lines 91-94.*

- Line 302 (and check throughout): don’t begin a sentence with this/it/they/etc. without clarifying what you are referring to

Thank you for point out.

Line 302, corrected to “These braarudosphaerid-enrichments thus suggest...”

Line 327 corrected to “The described strategy implies...”

- High-latitude restricted basins are often cooler than the latitudinal mean (Judd et al., 2020, GRL). This finding might help support your argument that the TEX₈₆-SST relationship is problematic in these settings

Thank you for pointing this out. Judd et al., (2020, GRL) indeed demonstrate the problem of SST calibration in specific regional settings such as restricted basins. To further emphasise this aspect, we have now added in line 218: “Notably, a recent study has also demonstrated that semi-enclosed basins document systematically offset from zonal mean temperatures, and are unusually warm and seasonal compared to the majority of open Ocean locations situated at the same latitudes, especially in the higher latitudes (Judd et al., 2020)”. This complements our point that caution must be taken when interpreting GDGT-TEX₈₆ and SST relationships.

- Restriction does not necessarily mean stratification. Do you have any geochemical or lithological data to support stratification of the water column?

We have additional low-resolution biomarker data, such as evidence of gammacerane and lycopane, which support water column stratification and bottom-water hypoxia. We decided not to include this information in the manuscript or the SI as the stratigraphic resolution is insufficient to confirm our

point. We plan to increase the resolution of these biomarker data and present them as a separate, follow on study.

- The depth of GDGT production may be assessed using the [2]/[3] ratio (Taylor et al., 2013, GPC). I suggest you investigate this geochemically if you are to make the claim that it affects your temperature reconstructions.

Thank you for suggesting this important and supporting observation of the [2]/[3] ratio, which we had included in our 'Source Data File' but not specifically discussed in the text. In lines 209-211, we now report this information: "moreover, we document a high GDGT-2/GDGT-3 ratio >5 in the study samples (on average 5.7; Supplementary Data Appendix), which also corroborates a contribution from Archaea living in the deeper water column (Taylor et al., 2013).

Reviewer #3 (Remarks to the Author):

Review of revised manuscript (June 2021)

Nature Communications manuscript NCOMMS-21-00870-A
by Cavalheiro et al. (2021)

- I am pleased to see the authors addressed my comments so comprehensively. Although the the TEX86 temperature calibration and its interpretation is still problematic, the according discussion is significantly improved. However, I am not confident with the reply that palynological data cannot provide quantitative estimates of paleotemperature. To my knowledge, the NLR approach allows to reconstruct mean annual as well as warmest month mean temperatures. Therefore I suggest to investigate at least a few samples of ODP 692 in respect to produce independent paleotemperature data for the Weissert Event.

The reviewer is correct in challenging our former statement, i.e. "palynological data cannot provide quantitative estimates of paleo-temperature changes but rather define the climatic boundary conditions on land". We now understand that our phrasing was misleading; in fact, we should have stated that after careful assessment of the literature, we cannot extract quantitative temperature estimates from the published data (Mohr et al., 1990).

This is because the data published are (1) not quantitative (counts for each taxon) but qualitative (presence/ absence) and (2) the sampling interval is too wide to add significant value to the discussion. We do not see how we can further use the published data to add value to our study. We concur with the reviewer that there are methods, such as the NLR approach, which allow for the reconstruction of mean annual as well as warmest month mean temperatures. Such interpretation, however, must rely on a rigorous statistical basis, otherwise results are meaningless or misleading (Grimm et al., 2016, RPP), especially for fossil assemblages as old as 130 Myr. Generating reliable data requires an appropriate volume of around 15g of bulk sediment for preparation for statistically sound palynological examination. Theoretically, this could have been a viable option we could have followed in the design of the study. We recognize, however, that additional palynological analyses are not feasible, because the material we have left is too limited. Resampling may lead to (possibly) a few horizons, randomly distributed, for palynology. We conclude that this approach is scientific flawed right from the outset and is therefore not favored in our study.

- Just a minor mistake:
Line 616. The name of the second author of ref 46 is Elling, not Felix.

Thank you for pointing out the mistake, now corrected.

REVIEWERS' COMMENTS

Reviewer #3 (Remarks to the Author):

The authors are correct that the original data set of Mohr (1990) does not allow for quantitative temperature estimates. However, I never meant to use the existing old palynological data, instead suggested to investigate new samples of ODP 692 in respect to produce independent paleotemperature data for the Weissert Event. 15 g is less than 10 ccm bulk sediment, which should be available in the same horizons of the GDGT samples. As noted in the first review, an independent temperature estimate would greatly increase the evidential value of the manuscript, given the uncertainty associated with the TEX86 temperature calibration in restricted ocean basins. But with this point of view I do not want to oppose the publication of the manuscript, which is - apart that - excellent.

Nature Communications point-by-point response to the reviewers' comments

We thank reviewer 3 for the continued support to improve our study and we deeply appreciate the conclusion that this study is 'excellent' and should be considered for publication in NC.

Any proxy approach using limited sample material, such as 10-20 cc from ODP, must prioritise which proxies to focus on. This implies a trade-off that not all excellent proxies can be measured on one sample. Our decision was on improved biostratigraphy, geochemistry and GDGT-SST, consuming the material we had to work with. The study outcome would not be possible without all three components. This was our choice and we firmly believe that this was excellent use of the material.

We fully agree that additional and independent temperature proxies would have been highly valuable to build more confidence in the TEX-SST record from ODP 692. Options would have been complementary t -reconstructions from e.g. palynomorphs, alkenone-precursors, oxygen isotopes, Ca/Mg, clumped isotopes, and others. This approach would certainly have improved the temperature story at ODP 692, which is a value on its own.

However, it would not have solved the wider issue of inter-comparability of different temperature proxy records from other published sections, globally. The solution would have been to apply a similar approach to all study sections worldwide to generate a global cross-calibrated temperature framework. This is obviously beyond any individual project and would have to be supported by a concerted effort from the wider research community. We welcome and encourage the reviewer to take an active and leading role in such a 'global' research effort, with a focus on palynology.

We actually make a different point in our study: by not 'a priori' favouring any specific proxy t -record (all with their individual limitations) but rather identifying the statistical best fit solution that combines all proxy, geological and model temperature reconstructions we offer an alternative strategy that deals with the issue of 'proxy limitations' and allows for further refinement/adjustment as new t data emerge.

We are confident that this new approach, applied to the Weisert Event as a case example, bears large merit that goes well beyond the example we have studied.

Yours sincerely,

Liyenne Cavalheiro and co-authors